# SILICON ISOTOPES OF DEEP-SEA SPONGES: NEW INSIGHTS INTO BIOMINERALISATION AND SKELETAL STRUCTURE

Lucie Cassarino[1], Christopher D. Coath[1], Joana R. Xavier[2, 3], and Katharine R. Hendry[1]

[1]University of Bristol, School of Earth Sciences, Wills Memorial Building, Queen's Road, Bristol, BS8 1RJ, UK
[2]CIIMAR - Interdisciplinary Centre of Marine and Environmental Research, University of Porto, Avenida General Norton de Matos, 4450-208 Matosinhos, Portugal
[3]Department of Biological Sciences and K.G. Jebsen Centre for Deep-Sea Research, University of Bergen, PO Box 7803, N-5020 Bergen, Norway

**Correspondence:** Lucie Cassarino (l.cassarino@bristol.ac.uk)

**Abstract.** The silicon isotopic composition ($\delta^{30}$Si) of deep-sea sponges skeletal element – spicules – reflects the silicic acid (DSi) concentration of their surrounding water, and can be used as natural archives of bottom water nutrients. In order to reconstruct the past silica cycle robustly, it is essential to better constrain the mechanisms of biosilicification, which are not yet well understood. Here, we show that the apparent isotopic fractionation ($\Delta^{30}$Si) during spicule formation in deep–sea sponges from the equatorial Atlantic ranges from -6.74 ‰ to -1.50 ‰ in relatively low DSi concentrations (15 to 35 $\mu$M). The wide range in isotopic composition highlights the potential difference in silicification mechanism between the two major classes, Demospongiae and Hexactinellida. We find the anomalies in the isotopic fractionation correlates with skeletal morphology, whereby fused framework structures, characterised by secondary silicification, exhibit extremely light $\delta^{30}$Si signatures compared with previous studies. Our results provide insight into the processes involved during silica deposition, and indicate that reliable reconstructions of past DSi can only be obtained using silicon isotope ratios derived from sponges with certain spicule types.

*Copyright statement.* TEXT

## 1 Introduction

### 1.1 Introduction to the Porifera world

Sponges (phylum Porifera) are one of the most primitive metazoans and have likely occupied ocean sea floors since the Precambrian period as indicated by molecular fossil records from the end of the Marinoan glaciation 635 Myr ago (Love et al., 2009) and Mongolian silica spicules dating from 545 millions years ago (Antcliffe et al., 2014). Sponges are obligate sessile organisms, most of which are efficient filter-feeders, capable of filtering 99% of the particles from water pumped through their internal body structure (Strehlow et al., 2017, and references therein). Most sponges secrete minerals such as calcite, aragonite and/or silica to build a complex and strong skeletal framework composed of elements called spicules, providing protection and

the maximum of contact between cells and their surrounding water (Uriz et al., 2003). Of the biomineralizing sponges, 92 % of living species produce silica, compared to 8% that produce calcium carbonate skeletons (Hooper and Van Soest, 2002) but this ratio may have varied in the past due to changes in paleo-ocean chemistry (Montanez, 2002) because sponges rely on the ion chemistry of their surrounding water to build their skeleton.

Three classes of sponges in the phylum Porifera, Homoscleromorpha, Demospongiae and Hexactinellida, produce their spicules made of bio-silica (amorphous silica) through the incorporation and deposition of hydrated silica ($SiO_2 \cdot nH_2O$), a process referred to as biosilicification (e.g. Uriz, 2006; Otzen, 2012). The spicules may represent up to 70–90% of the body (dry weight) depending on the species (e.g Sandford, 2003; Maldonado et al., 2012). Demosponges and hexactinellids differ in their body structure and spicule shape/size, which are both highly variable. Siliceous spicules can be subdivided into megascleres

(up to and beyond 300 $\mu$m) and microscleres (up to 50 $\mu$m), and are generally categorised by their size and their role in the skeletal framework (Uriz et al., 2003).

The Demospongiae is the largest class of the Phylum Porifera (Van Soest et al., 2012). Species within this class harbour monaxonic and/or tetraxonic megascleres with various shapes, as well as various types of microscleres, which compose their skeletal structure. Either Mega- and microscleres are loose, unfused, but joined by spongin (collagen protein) (Uriz, 2006).

Demospongiae have a cellular organisation i.e composed of cells that form tissues, which themselves form organs, which form an organism.

The Hexactinellida, commonly called glass sponges, exhibit a wide range of body structures, such as tubular, cup-shaped and branching (Ereskovsky, 2010). The spicules of hexactinellid sponges are characterised by hexactins (three axes with regular angles), which can lose or gain rays resulting in a wide range of shape and structure (Ereskovsky, 2010). One distinctive feature

of the Hexactinellida class is that their spicules can be loose, partially or totally fused, or even cemented by secondary silica layer and or junction (Uriz et al., 2003). One key feature that distinguishes between Demospongiae and Hexactinellida class is that Hexactinellida are characterised by a syncytial organisation, i.e. tissue composed of cells without individual plasma membrane (Leys and Lauzon, 1998; Maldonado and Riesgo, 2007).

Sponges have recently roused interest and are increasingly recognised as a key component of the marine silicon cycle

(Tréguer and De La Rocha, 2013; Maldonado et al., 2005). They live on the sea floor at most latitudes and depths (De La Rocha, 2003; Wille et al., 2010; Maldonado and Riesgo, 2007; Maldonado et al., 2012) and may be considered as a large living standing stock of silica in the oceans (Maldonado et al., 2010). Because of their relatively low growth rate and their inability to move, they are sensitive to change of their environment and because an individual sponge can live decades or centuries (Pansini and Pronzato, 1990; Leys and Lauzon, 1998) they can record information over long time periods (Jochum et al., 2017).

**1.2   Silicon isotope and deep sea sponges**

The silicon isotopic composition of biogenic silica ($\delta^{30}BSi$) has been introduced to study the past nutrient utilisation by De La Rocha et al. (1997) and since has been used to study the silicon cycle (e.g Hendry et al., 2016; Fontorbe et al., 2017). Silicon is composed of three stable isotopes, $^{28}Si$, $^{29}Si$, $^{30}Si$ with relative abundances of approximatively 92.23 %, 4.67 % and 3.10 % respectively (De Bièvre and Taylor, 1993). Silicon isotopic abundances in samples (SMP) are expressed as $\delta^{29}Si$ or $\delta^{30}Si$ with

the abundance ratio, $^{29}Si/^{28}Si$ or $^{30}Si/^{28}Si$ respectively and measured relative to the reference standard (NBS28). The results presented in this study are expressed as permil to be consistent with the International Union of Pure and Applied Chemistry (IUPAC) nomenclature. i.e.

$$\delta^x Si(\%o) = \left( \frac{\left(\frac{^x Si}{^{28}Si}\right)_{SMP}}{\left(\frac{^x Si}{^{28}Si}\right)_{NBS28}} - 1 \right) \tag{1}$$

The $\delta^{30}Si$ signature of deep sea sponges has been highlighted as a potential paleoceanographic proxy for silicic acid DSi concentration (De La Rocha, 2003; Hendry et al., 2010; Wille et al., 2010; Hendry and Robinson, 2012). The asymptotic relationship between DSi concentration and $\delta^{30}Si$ signature of sponge spicules ($\delta^{30}Si_{Spicules}$) is the result of the preferential incorporation of the lighter isotope (Wille et al., 2010; De La Rocha, 2003; Hendry et al., 2011). There is also a significant correlation between the apparent fractionation factor, $\Delta^{30}Si$ ($\delta^{30}Si_{Spicules}$ - $\delta^{30}Si_{DSi}$) and the ambient DSi concentration (Hendry and Robinson, 2012). The relationship between $\delta^{30}Si_{Spicules}$ and DSi concentration is not yet understood but a simple, biological model suggests that the fractionation factor could arise during sponge uptake, polymerisation and efflux (Wille et al., 2010; Hendry and Robinson, 2012). Thus, $\delta^{30}Si_{Spicules}$ is a potential proxy to quantify ocean changes in Si cycling with a larger spatial range and timescales than diatoms (De La Rocha, 2003; Fontorbe et al., 2017). Furthermore, the dissolution rate of sponge spicules is lower than diatoms frustules (Maldonado et al., 2005), which may result in a better preservation. A calibration of modern sponge specimens and core-top spicules from different oceans shows that the core-top specimens are not affected by post-depositional, dissolution or even early diagenesis (Hendry and Robinson, 2012), which are both potential concerns when dealing with the chemistry of reactive biogenic opal (Ragueneau et al., 2001; De La Rocha et al., 2011).

Despite the great potential of sponges as archives of past ocean Si cycling, there are still a number of outstanding questions relating to Si isotopic fractionation. Does Si isotopic fractionation remain constant during sponge growth? Can we trace DSi concentration over time by analysing $\delta^{30}Si$ sections of sponge skeleton? At what stage during biomineralisation does the Si isotopic fractionation occur, and does it vary with spicule morphology? Here, some of these questions are going to be addressed by investigating $\delta^{30}Si$ of modern deep-sea sponges collected from the equatorial Atlantic.

## 2 Material and Methods

### 2.1 Sample collection

Sponge samples were collected by remotely operated vehicle (ROV) at five stations, EBA, EBB, VEM, VAY and GRM between 298 m and 2985 m (figure 1) aboard the RRS James Cook on the TROPICS cruise (JC094), a West-East cross section in the equatorial Atlantic between ~5°N and ~15°N, from the 13[th] October to the 30[th] November 2013. Seawater samples were sampled using Niskin bottles attached to CTD rosette system and occasionally by ROV at each station. Whilst best attempts

were made to spatially match the sponge and water samples, it was not always possible to collect precisely co-located sponge and seawater samples. The $\delta^{30}Si_{DSi}$ values are reported in table A1 (appendix) and, for each sponge specimen, the closest seawater sample is used to calculate $\Delta^{30}Si$.

The sponge specimens were sampled either as large individual sponges or smaller individuals encrusted on other organisms such as corals. Subsamples were dried for transportation to UK.

Preliminary assignment of the specimens to the classes Demospongiae and Hexactinellida was carried out onboard using binocular and petrological microscopes. Identifications to lower taxonomic ranks, combining morphological and molecular methods, is underway and will be published in a separate paper.

## 2.2 Sponge spicules cleaning procedure and Si pre-concentration from seawater

For $\delta^{30}Si_{Spicules}$, organic matter (OM) was removed using hydrogen peroxide ($H_2O_2$, 30 % reagent grade). Subsamples of dry sponge specimens were transferred into 50 ml Eppendorf tubes, covered with $H_2O_2$ (30 % reagent grade) for 24 hours at room temperature then heated for 3 hours with new $H_2O_2$ (30 % reagent grade) at 85°C. The samples were rinsed with 18.2 M$\Omega$ Milli-Q water and heated for a further 3 hours with fresh $H_2O_2$, before a final Milli-Q rinse. Samples were transferred into clean Teflon vials to undergo further cleaning, 3 times in concentrated (16N) in-house Telfon-distilled $HNO_3$, rinsing between each stage with Milli-Q water. If remaining, lithogenic material was removed by hand before further cleaning steps. A subsample was taken and weighed before going through a final cleaning step. The subsample was covered with $HNO_3$ (16N Romil) and dried down at 120°C. When the spicules were dried, each sample was dissolved in 0.4M NaOH (Analar) at 100°C for 3 days, following published protocols (Ragueneau et al., 2005; Cardinal et al., 2007; Hendry and Robinson, 2012). Samples were acidified with 8N $HNO_3$ and diluted with Milli-Q water to reach pH 2–3. This cleaning procedure followed the technique in Hendry et al. (2010) and Hendry and Robinson (2012).

Prior to isotopic analysis of seawater, Si was pre-concentrated using the MAGIC method (MAGnesium-Induced Coprecipitation) of Karl and Tien (1992) with Reynolds et al. (2006) modifications. Brucite was precipitated overnight by the addition of 1.2 % v/v 1M NaOH. After centrifugation (3000 rpm for 3 minutes) the supernatant was transferred and 1 % v/v 1M NaOH was added and left overnight in order to extract any residual silicon. The two precipitates were combined and rinsed with 0.001M NaOH solution to remove remaining salt matrix before a final separation by centrifugation. Finally the precipitate, $Mg(OH)_2$, was dissolved by adding 8N $HNO_3$ resulting in a pH range of 1–3. The yield recovery of Si is equivalent to 92.1 %.

## 2.3 Analytical procedures

For $\delta^{30}Si$ analysis, pre-treated spicules and seawater samples were purified through cation ion exchange chromatography (Bio-Rad AG50W X12, 200-400 mesh in $H^+$ form). Analysis of $\delta^{30}Si_{Spicules}$ and $\delta^{30}Si_{DSi}$ were carried out by Multi-Collector Inductively-Coupled Plasma Mass Spectrometry (MC-ICP-MS, Finnigan Neptune s/n 1002) at the Bristol Isotope Group facilities, University of Bristol. All sample analyses were repeated at least twice and followed the typical standard-sample bracketing and Mg doping methods from Cardinal et al. (2003) with the best intensity match possible between samples and

bracketing standards. Measurement of secondary standards LMG-08 and Diatomite give $\delta^{30}$Si values of $-3.44 \pm 0.16$ ‰ (2 s.d., n = 173) and $1.23 \pm 0.15$ ‰ (2 s.d., n = 41), respectively. The external reproducibility of Si isotope measurements is $\pm 0.13$ ‰ and $\pm 0.17$ ‰ (2 s.d., degree of freedom = 212) for $\delta^{29}$Si and $\delta^{30}$Si respectively, where the analytical scatter for both standards has been pooled (Steele et al., 2012). For comparison, Hendry et al. (2011) and Reynolds et al. (2007) report $\delta^{30}$Si = $-3.37 \pm 0.17$ ‰ and $\delta^{30}$Si = $1.26 \pm 0.20$ ‰ for LMG-08 and Diatomite respectively. The new seawater standard ALOHA deep was analysed as an additional quality check, and yielded values within error of those obtained during an in-terlaboratory (Grasse et al., 2017): Aloha deep: $\delta^{30}$Si = $1.08 \pm 0.12$ ‰ and $\delta^{29}$Si = $0.58 \pm 0.12$ ‰ (2 s.d, n = 4). The $\delta^{29}$Si and $\delta^{30}$Si of all seawater and sponge samples is consistent with the kinetic mass fractionation law (Reynolds et al., 2007) i.e., $\delta^{29}$Si $vs.$ $\delta^{30}$Si has a slope of 0.516 (SE = 0.002, n = 362, $r^2$ = 0.995). The results are reported relative to the standard NBS28 (equation 1).

## 2.4 Scanning Electron Microscope images

Scanning Electron Microscope (SEM) images of sponge spicules have been carried out at the University of Bristol on a Hitachi S-3700N SEM. Clean spicules were sputter-coated with 10 nm of gold. The instrument was operating at an acceleration voltage of 15kV in second electron image mode.

## 3 Results

### 3.1 $\delta^{30}$Si of deep sea sponges

Data from the equatorial Atlantic exhibit $\delta^{30}$Si$_{\text{Spicules}}$ from $-5.51$ to $-0.51 \pm 0.13$ ‰ (2 s.d.) and $\Delta^{30}$Si from $-6.74$ to $-1.50 \pm 0.17$ ‰ (2 s.d.) (figure 2, a and b, respectively) representing the greatest $\Delta^{30}$Si observed in sponges to date (Hendry et al., 2010; Wille et al., 2010; Hendry and Robinson, 2012). Detailed results are presented in table A1 in the appendix. The results have been added to the existing calibration from Hendry et al. (2010); Wille et al. (2010); Hendry and Robinson (2012) (figure 2) showing that our new data are largely consistent with the existing calibration. However, a number of specimens deviate from the published calibration, and record unusually light isotopic signatures.

### 3.2 Degrees of spicule fusion

The SEM images from all sponges showing a $\Delta^{30}$Si larger than $-5$ ‰ and from 20 other samples chosen randomly within a lower $\Delta^{30}$Si range ($-1.50$‰ $< \Delta^{30}$Si $< -5$‰) have highlighted a variety of spicule shapes and degrees of skeletal fusion. In the bulk of sponge samples two groups can be identified: sponges with loose spicules with $\delta^{30}$Si$_{\text{Spicules}}$ following the published calibration curve, and sponges with fused spicules with $\delta^{30}$Si$_{\text{Spicules}}$ deviating from the published calibration curve.

Further, the SEM images of spicules with various $\delta^{30}$Si$_{\text{Spicules}}$ signature reveal five levels of fusion, defined here as F1, F2, F3, F4 and F5. Level F1 represents loose spicules, F2 spicules fused by node (netlike feature), F3 loose spicules fused in parallel with additional silica coating, F4 light dictyonal sketelon and F5 dense dictyonal skeleton, see figure 3 for SEM image

examples and table 1 for detailed description of the fusion degree from F1 to F5. It is observed that $\delta^{30}\text{Si}_{\text{Spicules}}$ and $\Delta^{30}\text{Si}$ show an enrichment in light isotopes with higher degree of spicule fusion (se figure 4).

## 4    Discussion

### 4.1    $\delta^{30}\text{Si}$ fractionation by deep-sea sponges

The new data presented here, from the equatorial Atlantic, show to date the largest range of $\delta^{30}\text{Si}_{\text{Spicules}}$ signatures and $\Delta^{30}\text{Si}$ for a small range in DSi concentration (15 to 35 $\mu$M). As $\Delta^{30}\text{Si}$ larger than $-5$ ‰, for this DSi range deviates from the published calibration curve, particular attention has been paid to these samples in order to understand the factors causing this large Si isotopic fractionation. SEM images show that these specimens have a common feature: a fused, dictyonal framework skeleton. The following discussion introduces in more detail the fractionation of Si isotopes by sponges and the hypotheses
relating to the large $\Delta^{30}\text{Si}$ from the dictyonal skeleton.

Previous studies tracking the $\delta^{30}\text{Si}$ of sponge silica have shown a non-linear relationship between $\delta^{30}\text{Si}_{\text{Spicules}}$ signatures and DSi concentration (Wille et al., 2010; Hendry and Robinson, 2012). The Si isotopic fractionation by sponges can be expressed either with $\Delta^{30}\text{Si}$ or $\varepsilon_f$ notation. Here $\Delta^{30}\text{Si}$ is defined by the difference between $\delta^{30}\text{Si}_{\text{Spicules}}$ and $\delta^{30}\text{Si}_{\text{DSi}}$, which describes
the observed apparent Si isotopic fractionation by sponges whereas $\varepsilon_f$ is the result from the biological model from Wille et al. (2010) (equation 2). Published data have shown $\Delta^{30}\text{Si}$ varying from -0.77 ‰ to -6.52 ‰ (figure 2b), which follow a non-linear relationship and cannot be described by a diatom-like Rayleigh fractionation (characterised by a constant fractionation factor during DSi utilisation) because isotopic fractionation during the uptake of DSi by sponges is variable, increasing with DSi concentration. Wille et al. (2010) proposed a model following Milligan et al. (2004), which suggests that Si isotopic
fractionation in marine sponges is mainly controlled by Si uptake. Reincke and Barthel (1997) first investigated the formation of BSi (i.e. silicification) in cultured sponges by regeneration of sponges pieces and, more recently, Si uptake has been investigated in culture using whole sponges collected at sea that were then transferred to a controlled environment (Maldonado et al., 2011; López-Acosta et al., 2016, 2018). Despite the different set-up and species chosen for each experiment, all culture experiments carried out to date suggest that the silicification in sponges is controlled by enzymatic processes, exhibiting Michaelis-Menten
enzyme kinetics, and is dependent on substrate concentration, here DSi. From the close resemblance of the DSi and $\delta^{30}\text{Si}_{\text{Spicules}}$ relationship and the growth rate kinetics, Wille et al. (2010) proposed a model from which $\delta^{30}\text{Si}$ is fractionated during the

uptake phase and internal spicule formation. The related fractionation of Si isotopes is expressed as $\varepsilon_f$ (equation 2), with DSi concentration the main factor influencing $\delta^{30}\mathrm{Si}_{\mathrm{Spicules}}$.

$$\varepsilon_f = \varepsilon_{tI} + (\varepsilon_p - \varepsilon_E)\left\{1 - \frac{\dfrac{V_{\mathrm{max,P}}}{\left(\dfrac{K_{\mathrm{m,P}}}{\mathrm{DSi}}\right)+1}}{\dfrac{V_{\mathrm{max,I}}}{\left(\dfrac{K_{\mathrm{m,I}}}{\mathrm{DSi}}\right)+1}}\right\} \tag{2}$$

where $\varepsilon_{tI}$ is the Si isotopic fractionation during Si uptake, $\varepsilon_p$ is the Si isotopic fractionation during polymerisation, $\varepsilon_E$ is the Si isotopic fractionation during the efflux, $V_{\mathrm{max,P}}$ and $V_{\mathrm{max,I}}$ are the maximum polymerisation and incorporation rates, respectively, $K_{\mathrm{m,P}}$ and $K_{\mathrm{m,I}}$ are the half saturation constant of polymerisation and incorporation respectively and DSi the silicic acid concentration of the surrounding water.

Hendry and Robinson (2012) applied this model to a wide range of modern sponges from different ocean basins showing that the temperature, one of the factors controlling enzymatic processes, does not affect the relationship between $\Delta^{30}\mathrm{Si}$ and DSi concentration, which supports DSi concentration being the main factor controlling Si isotopic fractionation. Despite the small range of seawater temperature in this study, our data show no relationship between $\Delta^{30}\mathrm{Si}$ and temperature (figure A1 in appendix). However, here a group of sponges from the equatorial Atlantic exhibit a different relationship between $\Delta^{30}\mathrm{Si}$ and DSi concentration with a very large apparent fractionation, $\Delta^{30}\mathrm{Si} < -5\,‰$, at low concentration (15 to 35 $\mu$M). Figure 4a shows the variability of $\Delta^{30}\mathrm{Si}$ for each fusion stage, and that the fusion degree of the spicules appears to affect $\Delta^{30}\mathrm{Si}$. Furthermore figure 4b shows the $\Delta^{30}\mathrm{Si}$ residual of the same samples against the previous published calibration (Hendry et al., 2010; Wille et al., 2010; Hendry and Robinson, 2012), which suggests that other processes than silicic acid concentration are involved in the fractionation of Si isotopes.

Dictyonal framework skeletons, F4 and F5, only belong to the Hexactinellida class, which could suggest that the two classes have a different Si isotopic fractionation due to their different silicification mechanism (Maldonado and Riesgo, 2007). The $\delta^{30}\mathrm{Si}_{\mathrm{Spicules}}$ average signature of the two siliceous sponge families from the compiled data presented in figure 2 (Hendry and Robinson (2012), Wille et al. (2010), Hendry et al. (2010) with the equatorial Atlantic data (JC094)) show that the Hexactinellida class is significantly lighter than the Demospongiae, with $\delta^{30}\mathrm{Si}_{\mathrm{Spicules}} = -2.66 \pm 0.21\,‰$ (C.I. of the mean) and $-1.91 \pm 0.30\,‰$ (C.I. of mean) respectively. However, it is important to take into consideration the environmental conditions of growth because $\delta^{30}\mathrm{Si}_{\mathrm{Spicules}}$ depends on the $\delta^{30}\mathrm{Si}$ and DSi concentration of seawater and the two groups live at different depth ranges and nutrient conditions. To eliminate the influence of these two parameters and resolve whether or not Demospongiae and Hexactinellida fractionate Si isotopes in different ways, a $\Delta^{30}\mathrm{Si}$ residual has been calculated. $\Delta^{30}\mathrm{Si}$ residual = $\Delta^{30}\mathrm{Si}_{\mathrm{Observed}} - \Delta^{30}\mathrm{Si}_{\mathrm{Spicules\ best\ fit}}$. Three best fits have been calculated assuming the hyperbolic relationship between DSi and

$\Delta^{30}$Si (Hendry and Robinson, 2012) to deconvolve the influence of fused skeleton. Equation 3, 4 and 5 correspond to the best fit curves (number in parentheses are the standard error) in figure 5, for:

1) the previous compilation from Hendry and Robinson (2012), Wille et al. (2010), Hendry et al. (2010),

$$\Delta^{30}\text{Si} = -5.39(0.4) + 111.51(11.3)/(26.87(11.2) + \text{DSi}) \tag{3}$$

2) the previous compilation from Hendry and Robinson (2012), Wille et al. (2010), Hendry et al. (2010) + this study data,

$$\Delta^{30}\text{Si} = -4.69(0.2) + 30.89(2.7)/(7.29(2.7) + \text{DSi}) \tag{4}$$

3) the previous compilation from Hendry and Robinson (2012), Wille et al. (2010), Hendry et al. (2010) + this study without the dyctional skeleton,

$$\Delta^{30}\text{Si} = -4.71(0.18) + 34.25(2.6)/(8.29(2.6) + \text{DSi}) \tag{5}$$

Figures 5b, d, and f show $\Delta^{30}$Si residual results of each class, calculated from the best fit (figure 5a,c, e, respectively) with, a) the compilation of published data (Hendry and Robinson, 2012; Wille et al., 2010; Hendry et al., 2010), b) of published and all JC094 data, and c) of published and JC094 data without the fused spicules (F1 to F5). Only the samples with class identification are include in the three residual results. The results of these residual tests show that there is no disparity between the two classes even with the incorporation of the dyctional framework (figure 5b, d, f). The residual test, on the other hand, highlights that hexactinellids have a tendency to live in water with higher DSi concentration compared to demosponges, which supports the idea that the Si isotopic fractionation is driven by DSi concentration. Furthermore, when data from all JC094 data are incorporated into the published calibration curve (figure 5c), the dictyonal framework (F4 and F5) are not included in the 95 % confidence limits (red lines). This observation illustrates that sponges with fused spicules, in particular dictional framework F4 and F5, cannot yet be use as a robust proxy for ocean chemistry.

The question still remains as to what controls the large $\Delta^{30}$Si observed for sponges with complex dictyonal framework skeletons. Two main hypotheses are proposed and discussed in order to deconvolve $\delta^{30}\text{Si}_{\text{Spicules}}$ and fusion type.

## 4.2 Spicule composition: a control of $\Delta^{30}$Si?

The primary hypothesis concerns the structure of the spicule itself. Recently He et al. (2016) have shown, using chemical modelling, that there is an extremely large Si isotopic fractionation of -9.1 ‰ at 25°C between hyper-coordinated organosilicon complexes and DSi. This paper has raised the idea that the organic content inside the spicule itself could impact the fractionation of Si isotopes during biosilicification.

A spicule is composed of hydrated amorphous silica $(\text{SiO}_2)_{2-5}.\text{H}_2\text{O}$ with Si and O up to 75 % and more, and 6–13 % of water, with some traces of other elements (Sandford, 2003; Schröder et al., 2008) and organic molecules (Uriz et al., 2003). Biosilicification is mediated by enzymes such as silicatein during the formation of the spicule, where silica layers are deposited around an organic axial filament containing the mature silicatein (Cha et al., 1999; Müller et al., 2008; Wang et al., 2012a). The

spicule formation starts with an immature spicule inside a sclerocyte, and Si is supplied by internal vesicles, the silicasomes. The immature spicule is extruded by evagination from the sclerocyte, resulting in an axial elongation. In the extracellular space the elongated immature spicule is in contact with silicatein and galectin (protein with structural function), which mediate the deposition of silica released from external silicasome vesicules (Müller et al., 2013). One major difference between hexactinel-
lids and demosponges is that the spicules in demosponges fuse their silica and organic layers, constituting the primary spicule, when extruded from the sclerocyte (Müller et al., 2008; Wang et al., 2011, 2012a) while the concentric silica layers could remain separated with thin organic layers in hexactinellids (Aizenberg, 2005; Müller et al., 2009; Wang et al., 2012b).

Thermal analysis showed that the hexactinellid *Scolymastra joubini* spicules are composed of 15 % OM compared to demo-sponges with 10 % (Croce et al., 2004), Furthermore, SDS-PAGE analysis of Hexactinellid *Euplectella aspergilum* has shown
that the proteins of its axial filament have higher molecular weights than those isolated in demosponges (Weaver and Morse, 2003), which supports the difference in organic content between the two classes. The larger Si isotopic fractionation of sponges with a dyctional framework could be a result of a much greater number of organosilicon complexes within the structure. In-deed, Weaver et al. (2007) showed by SEM that the internal skeletal structure of the hexactinellid sponge *E. aspergillum* is comprised of small spicules that are embedded in a silica matrix surrounding a larger spicule (figure B1 a), in appendix). The
structural dictyonal framework consists then of multiple layers of silica/organic composite (figure B1c) and d), in appendix).

Nevertheless, results from the residual tests (figure 5) show that there is no significant difference in $\Delta^{30}$Si between Hex-actinellida and Demospongiae classes despite the difference in their spicule composition. This suggests that the large $\Delta^{30}$Si in sponges that display a dictyonal framework is not solely a result of the differences in organic composition of the spicules but could also be controlled by the enzymes that mediate silica deposition.

## 4.3 An enzymatic control of $\Delta^{30}$Si?

The second hypothesis relates to the growth rate kinetics of the sponges. As proposed by Wille et al. (2010) the fractionation of Si isotopes by sponges, $\varepsilon_f$ (see equation 2). The Si isotopic fractionation by sponges is assumed to occur during Si up-take and during internal spicule formation with spicule formation being a function of Si influx and efflux from the sclerocyte (Milligan et al., 2004). The efflux is the difference between Si incorporated into the sclerocyte and Si used to form the spicule
(i.e. polymerisation) (Wille et al., 2010, and references therein). To date, only Reincke and Barthel (1997); Maldonado et al. (2011); López-Acosta et al. (2016) and López-Acosta et al. (2018) have cultured sponges to investigate the Michaelis-Menten enzyme kinetics of sponges. In this section, $\varepsilon_f$ has been modelled using $K_{m,P}$ and $V_{max,p}$ values from these four sponge culture experiments and are summarised in table 2.

In order to compare the effect of the kinetic parameters on Si isotopic fractionation following the Wille et al. (2010) model (equation 2), the six simulations have been undertaken with a constant value for $\varepsilon_p - \varepsilon_E$ corresponding to $-5.39$ ‰. $\varepsilon_p - \varepsilon_E$ was calculated from the hyperbolic relationship between DSi and $\Delta^{30}$Si following equation 3 for each sponge species from the culture studies. $V_{max,I}$ and $\varepsilon_{tI}$ are the constant values defined by the minimum misfit function described in Wille et al. (2010), with $V_{max,I} = 120$ $\mu$mol Si h$^{-1}$ g$^{-1}$ and $\varepsilon_{tI}$ = -1.34 ‰.

Figure 6 shows the results of $\varepsilon_f$ simulated from the six kinetic parameters detailed above and in table 2. The results of this simulation show lighter $\varepsilon_f$ from *Haliclona simulans* to *Tethya citrina*. This observation coincides with the $K_{m,P}$ values of each species where $K_{m,P}$ of *Haliclona simulans* is the highest and $K_{m,P}$ of *Tethya citrina* is the smallest (table 2). $K_{m,P}$ value illustrates the affinity of an enzyme with a substrate, i.e. the smaller $K_{m,P}$, the higher the affinity between the substrate (DSi) concentration and the enzyme. In other words, $K_{m,P}$ informs about the binding efficiency between the substrate and the enzyme sites and is used as an indicator of competitiveness between species. The comparison of the six species suggest that the lower the $K_{m,P}$, the larger is $\varepsilon_f$ and $\varepsilon_f$ calculated from López-Acosta et al. (2016)$_{T.citrina\ sp}$ is related to *Tethya citrina* sp., which has the lowest $K_{m,P}$ value, showing a high affinity for DSi compare to the five other species. Unfortunately, to date there are no published culture studies related to the class Hexactinellida but it is likely that their $K_{m,P}$ show higher affinity with DSi due to their high requirements for silicon.

In López-Acosta et al. (2016), *Tethya citrina* is more silicified than *Hymeniacidon perlevis* and have a twice higher $V_{max,p}$ and 3 times lower $K_{m,P}$. Dictyonal frameworks display very dense skeletons compared to the demosponges made of loose spicules. Here, the hypothesis made is that the affinity between the enzyme and DSi is higher for sponges with a dictyonal framework, which means that their $K_{m,P}$ value is likely to be lower. The simulation of Si isotopic fractionation has been made using the *T. citrina* sp kinetic parameters except $K_{m,P}$ that has been reduced to 10 $\mu$M. Results of this simulation is referred to as Low $K_m$ in figure 6. It is clear that the reduction of the half saturation constant ($K_{m,P}$) value has a major impact on the Si isotopic fractionation with larger Si isotopic fractionation specifically for the low concentration, where DSi = 20–30 $\mu$M (figure 6). However, the very large $\Delta^{30}$Si of -6.74 ‰ for DSi = 20.05$\mu$M could be modelled by decreasing $K_{m,P}$ to 1 $\mu$M. Whilst this value of $K_{m,P}$ is low, and potentially not biologically plausible, more research is required in order to constrain enzymatic control in hexactinellid biomineralization. However, a low $K_m$ value suggests that the sponges need only a small amount of substrate (DSi) to grow, which means that sponges with a low $K_m$ are likely to grow in seawater with a low DSi concentration. When plotted against depth, the dictyonal framework skeletons, level F5, are located between 1100 m and 1800 m (see figure 7) corresponding to a regional minimum in DSi concentration as shown in figure 7b and 1b. This observation reinforces the hypothesis that the kinetic parameter $K_{m,P}$ is involved in Si isotopic fractionation. However, to successfully model the highest $\Delta^{30}$Si (-6.74 ‰) we must also investigate the potential impact of efflux on Si isotopic fractionation.

Biosilicification in sponges results in the condensation of DSi into BSi controlled by silicifying enzymes, such as silicatein (Cha et al., 1999; Müller et al., 2008), which interacts with other enzymes and proteins, for example galectin and collagen (Krasko et al., 2000; Müller et al., 2013) to form their spicule. The bonding reactions made by silicatein during spicule formation appear to be reversible. For example, the spicule formation of the Demospongiae *Suberites domuncula and Geodia cydonium*, is the result of an anabolic reactions (bonds being created) via silicatein and catabolic reactions (breaking of bonds) via silicase (Müller et al., 2008). This dynamic feedback regulates the internal cell DSi concentration (Wang et al., 2012b), which could suggest that the efflux from the sclerocyte is more important than previously thought. To test this hypothesis, $\varepsilon_f$ has been simulated with $K_m$ = 10 $\mu$M (same as Low $K_{m,P}$ presented earlier) and by increasing the $\delta^{30}$Si fractionation from the efflux of Si from the sclerocyte, $\varepsilon_E$, to 5 ‰. The result presented in figure 6 (referred to as Low $K_m$ + efflux) shows that the largest observed $\Delta^{30}$Si value of -6.74 ‰ can be modelled by increasing $\varepsilon_E$. Our results highlight the possibility that sponges

that grow in a low DSi environment are "tuned" to take up DSi rapidly into their sclerocytes, releasing any excess via efflux processes. Whilst further work is required to understand sponge biomineralisation, our results show the potential for Si isotopes as a tool to investigate cellular uptake and silicification processes.

## 4.4 Comparison with previous studies

The new data set of this study show a wide range of $\Delta^{30}$Si for a small range of DSi concentration compared to previous studies (figure 2) (Hendry and Robinson, 2012; Wille et al., 2010; Hendry et al., 2010). Here spicule shape, in particular the fusion stage, and $\Delta^{30}$Si have been investigated and are closely related, where high fusion stages show very large $\Delta^{30}$Si, which deviate from the existing calibration. Why has this relationship between spicule fusion and $\Delta^{30}$Si not been observed in previous studies? This new data set is composed of 103 samples in which 15 are deviating from the calibration and display a dyctional

skeleton. Previous studies are based on fewer samples and all the hexactinellid specimens have been found, except for one, in high DSi environments (higher than 45 $\mu$M) (Hendry and Robinson, 2012; Wille et al., 2010; Hendry et al., 2010). As the spicules in Hexactinellida class can be loose, partially or totally fused, or even cemented by secondary silica (Uriz et al., 2003), it is likely that previous studies only analysed samples with loose spicules (equivalent to F1 here). Furthermore, Hendry and Robinson (2012) piblished one sample with $\Delta^{30}$Si = -6.52 ‰ for DSi = 125 $\mu$M (figure 2). This sample also displayed a fused

skeleton but at this date the large fractionation was attributed to the lack of constraint on ambient seawater $\delta^{30}$Si .

## 5  Conclusions

Marine sponges are potential geochemical archives of present and past oceanic silicon cycling. Through a simple kinetic model it is possible to predict $\delta^{30}$Si fractionation of modern sponges, which support the use of Si isotopes in the reconstruction of past silicic acid concentration of bottom waters. However, the data presented here illustrate that the proxy has its limits. The

skeleton type and, in particular, the level of fusion of the skeleton lattice impacts the Si isotopic fractionation significantly. Sponges displaying a dictyonal framework do not fit the asymptotic relationship with DSi observed in previous studies. This divergence also has been observed for a carnivorous sponge (Hendry et al., 2015) where it is suggested that the Si isotopic fractionation is associated with a specific hypersilicified spicule type (desma). Here, we suggest that the organic template responsible for spicule formation, the organic matter content that differs between the two major classes, could influence the Si

isotopic fractionation. However, residual tests have shown that there is no significant difference between Hexactinellida and Demospongiae classes when differences in habitat and nutrient conditions are taken into consideration. Yet, this study shows a strong relationship between enzyme kinetic parameters ($K_m$) and $\varepsilon_f$, with low $K_m$ related to lighter Si isotopic fractionation, which suggests that Si isotopic fractionation is dependent on the equilibrium between molecules and bonding interactions mediated by catabolic and anabolic reactions in the process of biomineralisation. As yet, sponge biomineralisation processes

are not fully understood and further work is required to understand the specific pathways involved, especially in the case of the hexactinellids.

*Sample availability.* Sample and sample images are available at the University of Bristol, for further detail contact Katharine R. Hendry, email address: k.hendry@bristol.ac.uk

*Author contributions.* LC, conducted the analysis, created the figures and wrote the paper. CDC, helped conducting the isotopic analysis. KRH, conducted the fieldwork and conceived the study. JRX, identified the samples. All authors have reviewed the manuscript.

5 *Competing interests.* The authors declare that they have no conflict of interest.

*Acknowledgements.* We acknowledge the science team and the crew of JC094 and Laura Robinson for cruise organisation. We also would like to thank Paul Curnow for constructive comments, Stuart Kearns for his SEM training and assistance, and Maria López-Acosta for her help. Finally the funding from the Royal society (Grant code RG130386) and from The European Research Council. Joana R. Xavier received support from the European Union's Horizon 2020 research and innovation program through the SponGES project (grant agreement
10 No. 679849)

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

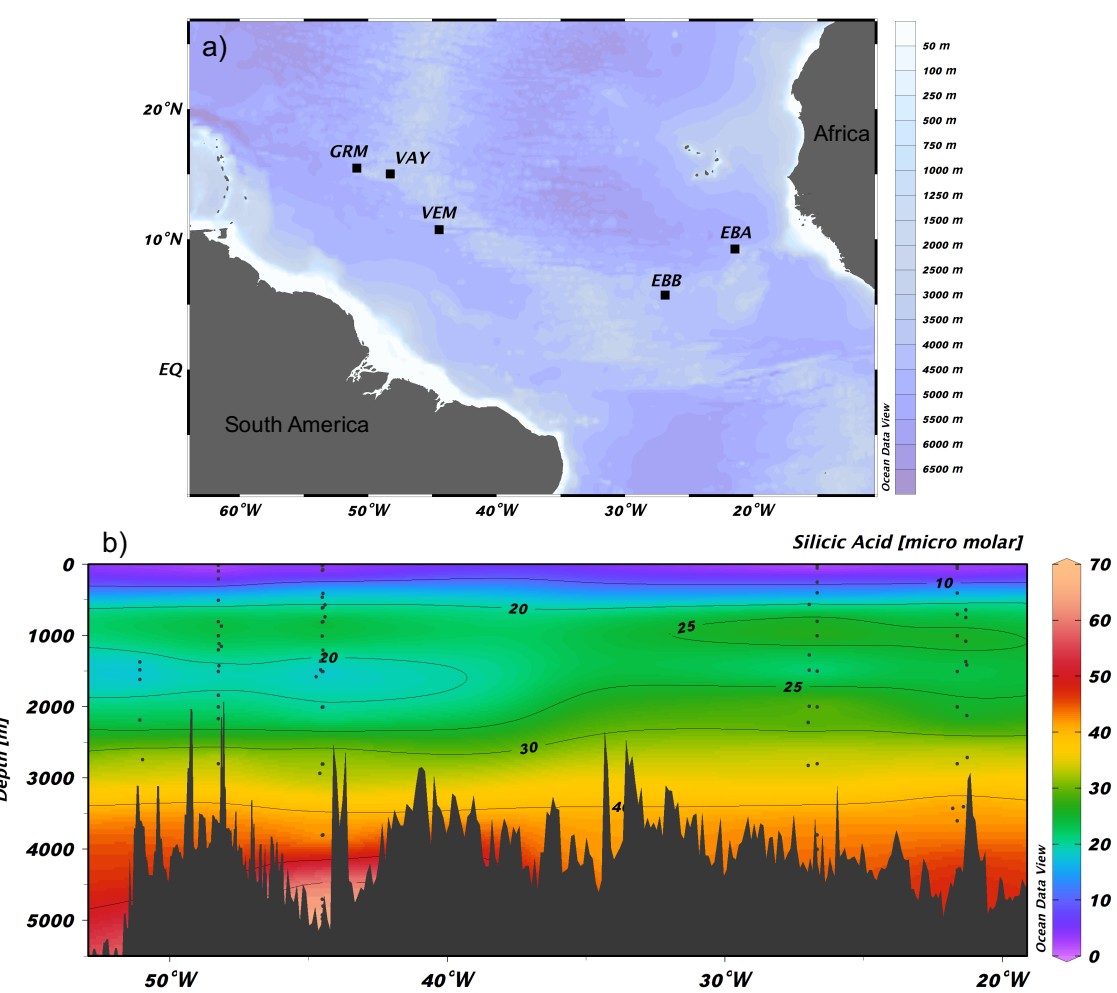

**Figure 1.** a) JC094 sampling stations from the equatorial Atlantic. From east to west: EBA (Carter Seamount), EBB (Knipovich Seamount), VEM (Vema Fracture Zone), VAY (Vayda Seamount), GRM (Gramberg Seamount). b) Cross-section of DSi concentration in micro molar along the JC094 transect. Dots representing the sampling depth of stations, from Est to West: EBA, EBB, VEM, VAY and GRM.

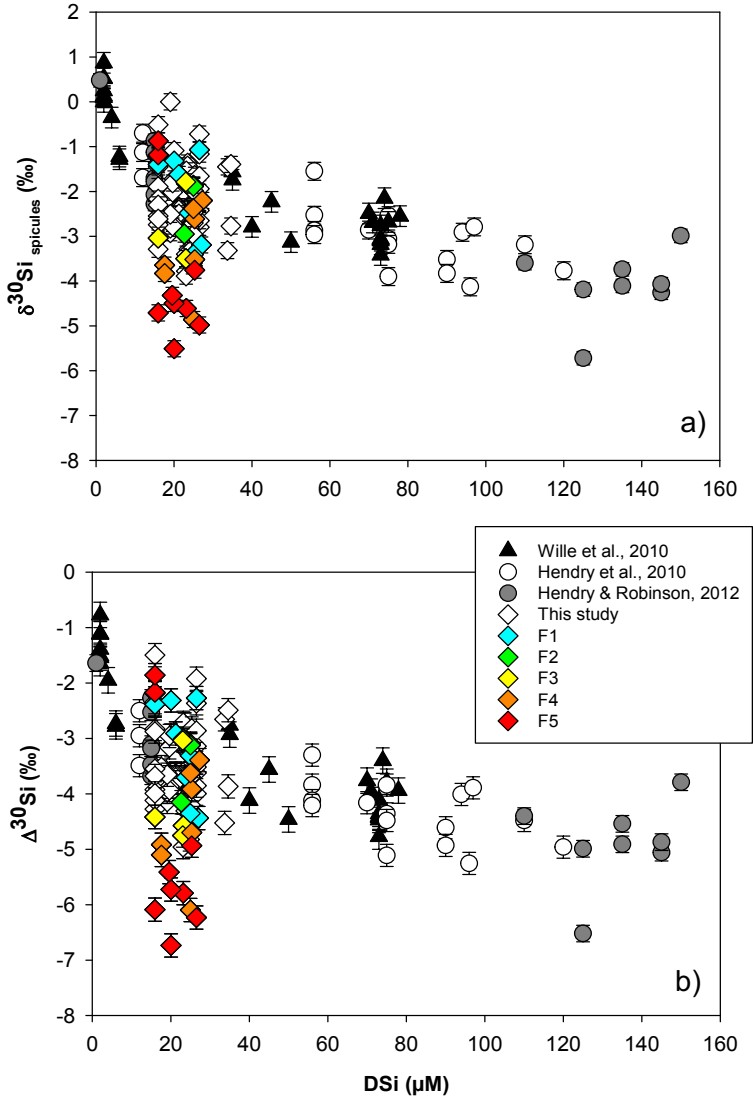

**Figure 2.** Sponge calibration data from Hendry et al. (2010) (Drake passage and Scotia sea), Wille et al. (2010) (Antarctica, Tasmania and New Zealand), Hendry and Robinson (2012) (North Atlantic, West Antarctic Pensinsula, Woods Hole and North Pacific) and data from this study (equatorial Atlantic, JC094). a) Silicon isotopic composition of the spicules ($\delta^{30}Si_{Spicules}$) and b) deep sea sponges apparent Si isotopic fractionation ($\Delta^{30}Si$) against DSi concentration. Error bars for this study are showing 2 s.d, $\pm$ 0.13 for $\delta^{30}Si_{Spicules}$ and $\pm$ 0.17 for $\Delta^{30}Si$. NB: core top samples from Hendry and Robinson (2012) are not included.

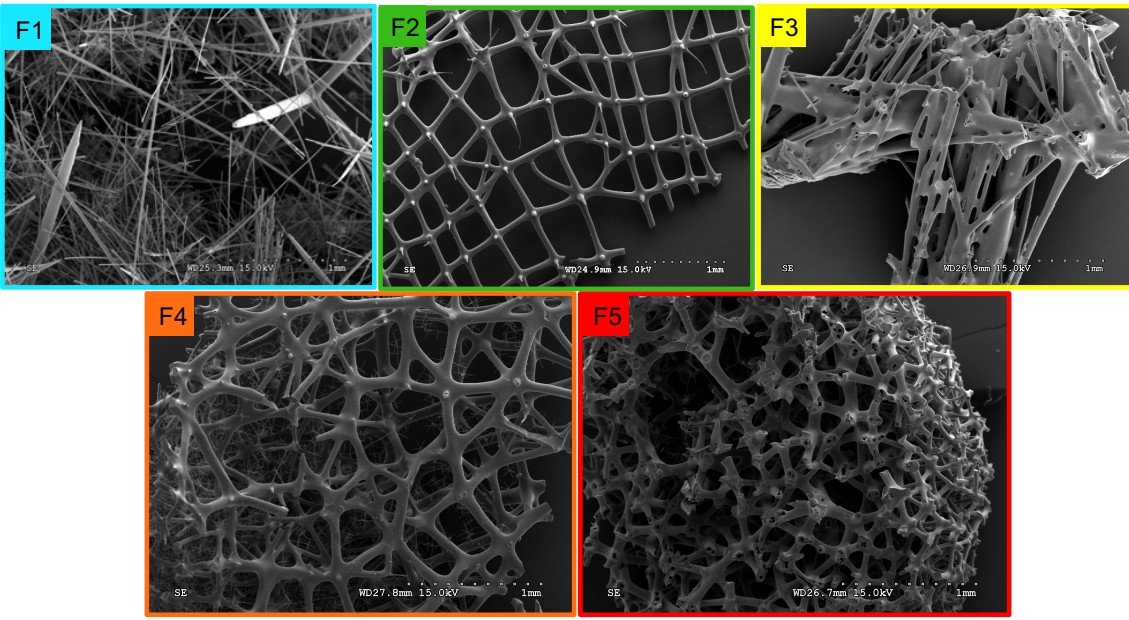

**Figure 3.** Level of fusion of sponge spicules from the equatorial Atlantic. F1 (blue) loose spicule, F2 (green) net-like, F3 (yellow) additional silica coating, F4 (orange) light dictyonal sketelon and F5 (red) dense dictyonal skeleton.

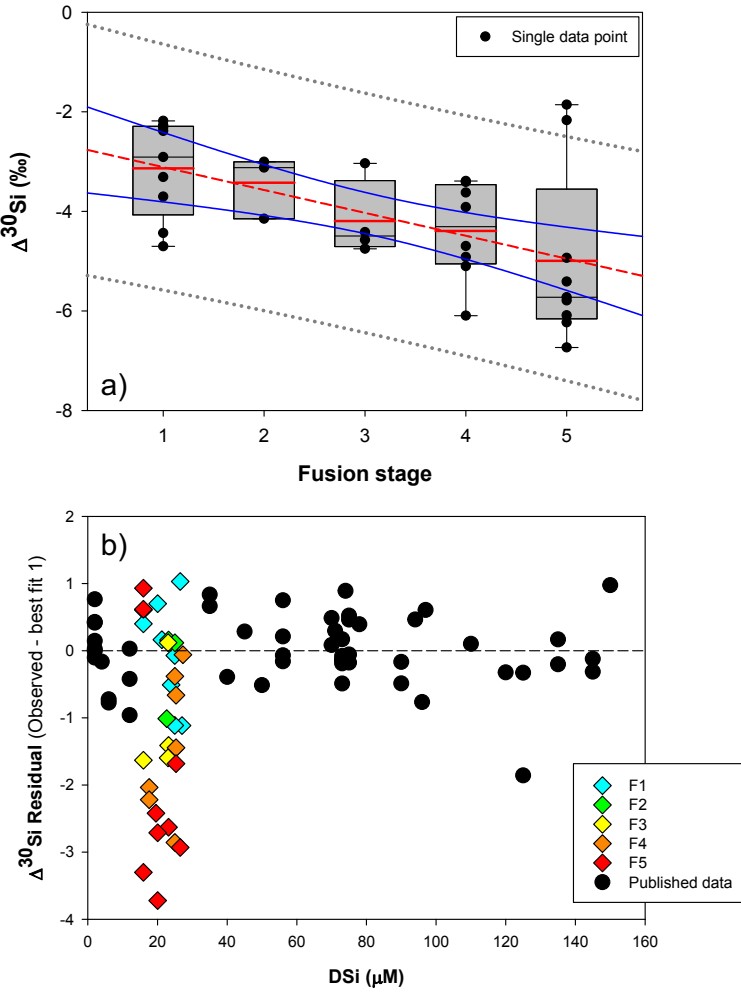

**Figure 4. a)** Boxplot showing the apparent Si isotopic fractionation $\Delta^{30}Si$ by sponges as a function of spicule fusion degree, 1 for loose spicules (F1), 2 for net-like (F2), 3 for additional silica coating (F3), 4 for dictyonal framework (F4) and 5 for dense dictyonal framework (F5). Red lines in the boxplots are the mean of each population and black the median. The box define the 25[th] and the 75[th] percentiles and the error bars are 10[th] and 90[th] percentile. Black circles show single sample for each of the fusion stage and the red dashed line shows the significant linear relationship between $\Delta^{30}Si$ and the fusion stage of the spicule (n = 33, p<0.0011). The blue lines represent the 95% confidence band and the grey short dashed lines the 95% prediction band. Data plotted here correspond to the samples from the equatorial Atlantic with identified fusion stage i.e. coloured diamond of b) and so occupy a very narrow range of DSi. **b)** $\Delta^{30}Si$ residual of samples with identified fusion stage compared to the published calibration (best fit 1).

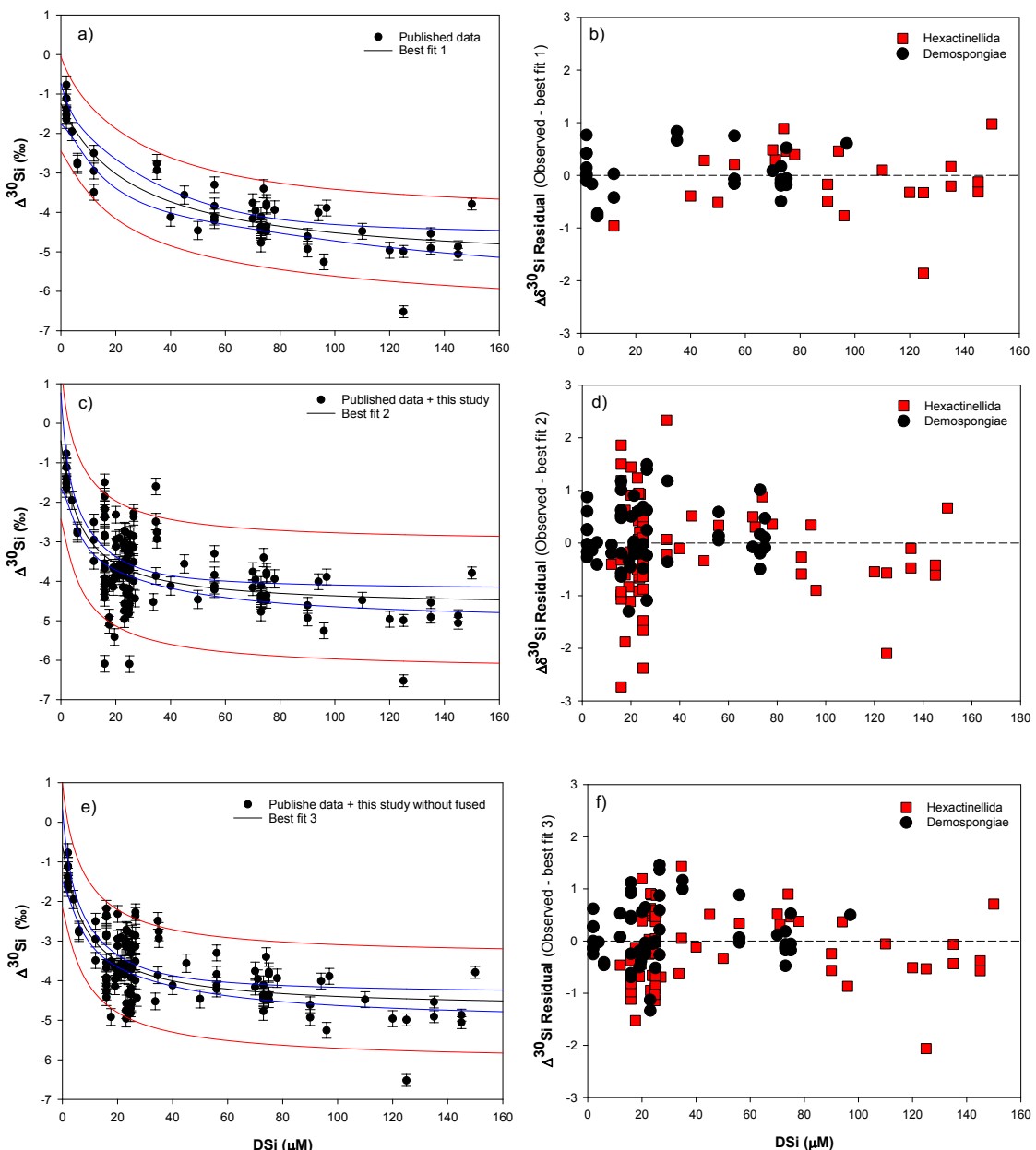

**Figure 5.** Plots of the best fit model (left) and the corresponding $\Delta^{30}Si$ residual (right) of Hexactinellida (red squares) and Demospongiae (black circles) class. a) Best fit ($\Delta^{30}Si = -5.39(0.4) + 111.51(11.3) / (26.87(11.2) + DSi)$) and b) Residual from Hendry and Robinson (2012), Wille et al. (2010), Hendry et al. (2010) data. c) Best fit ($\Delta^{30}Si = -4.69(0.2) + 30.89(2.7) / (7.29(2.7) + DSi)$) and d) Residual from Hendry and Robinson (2012), Wille et al. (2010), Hendry et al. (2010) with the equatorial Atlantic data (JC094). e) Best fit ($\Delta^{30}Si = -4.71(0.18) + 34.25(2.6) / (8.29(2.6) + DSi)$) and f) Residual from Hendry and Robinson (2012), Wille et al. (2010), Hendry et al. (2010) with the equatorial Atlantic (JC094) without the fused spicules (F2 to F5). For best fit model plots, black lines = best fit regression, blue lines = 95% confidence interval and red lines = 95% prediction interval and for best fit equation number in parentheses are the standard error. NB, the best fit 2 and 3 and their corresponding residual test were calculated only with identified sponge samples, which results in discrepancies between figure 2b and 6.

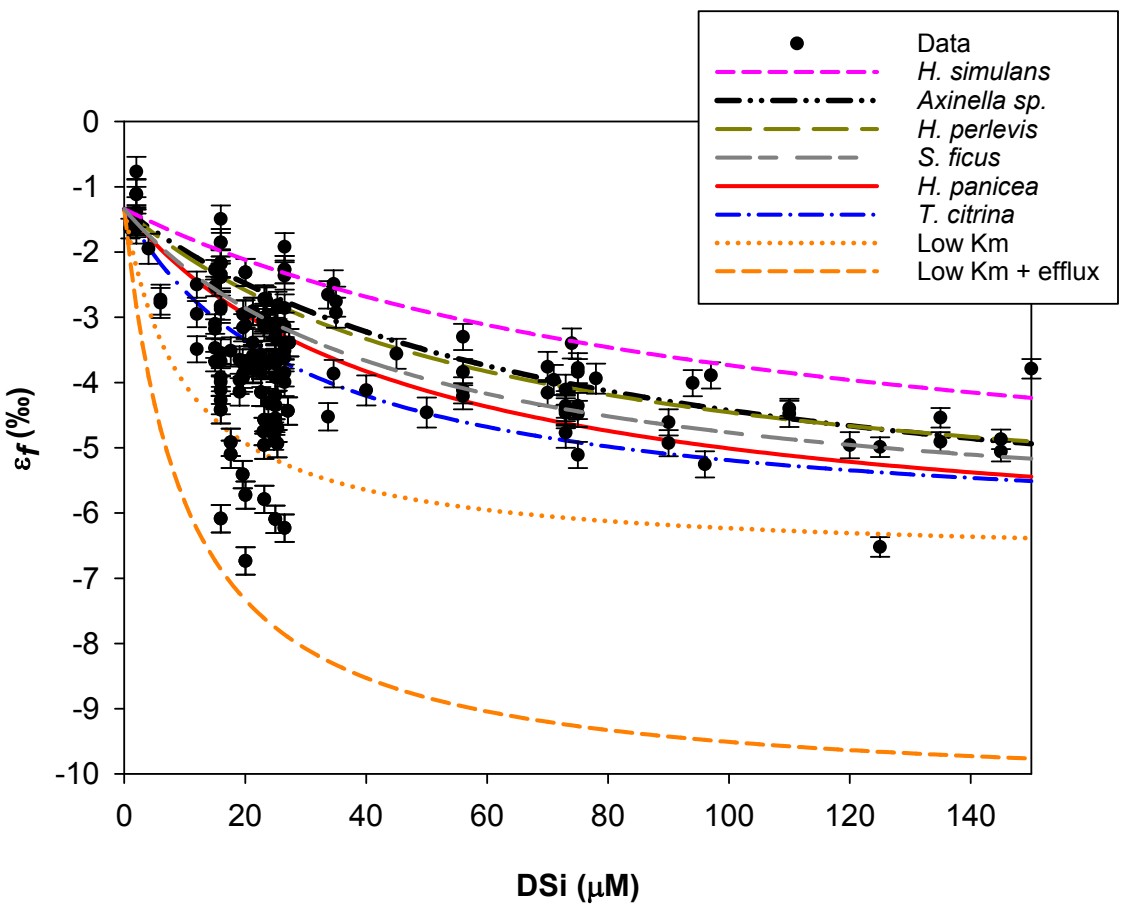

**Figure 6.** Michaelis-Menten kinetic model of Si isotopic fractionation (equation 2) for six parameterisations (see table 2).Black circles are all previous published and JC094 data. Sponge species from top to bottom: Pink: $\varepsilon_f$ from *Haliclona simulans* kinetic parameters López-Acosta et al. (2018), Black: $\varepsilon_f$ from *Axinella* sp. (Maldonado et al., 2011), Green: $\varepsilon_f$ from *Hymeniacidon perlevis* López-Acosta et al. (2016), Grey: $\varepsilon_f$ from *Suberites ficus* López-Acosta et al. (2018), Red: $\varepsilon_f$ from *Halichondria panicea* (Reincke and Barthel, 1997), Blue: $\varepsilon_f$ from *Tethya citrina* (López-Acosta et al., 2016). The following curves are experimental tests, Orange: $V_{max,p}$ of (López-Acosta et al., 2016)$_{T.citrina\ sp}$ and $K_m$ (10 $\mu$M), with $\varepsilon_E$ as 1.39 (dotted) or 5 ‰ (dashed).

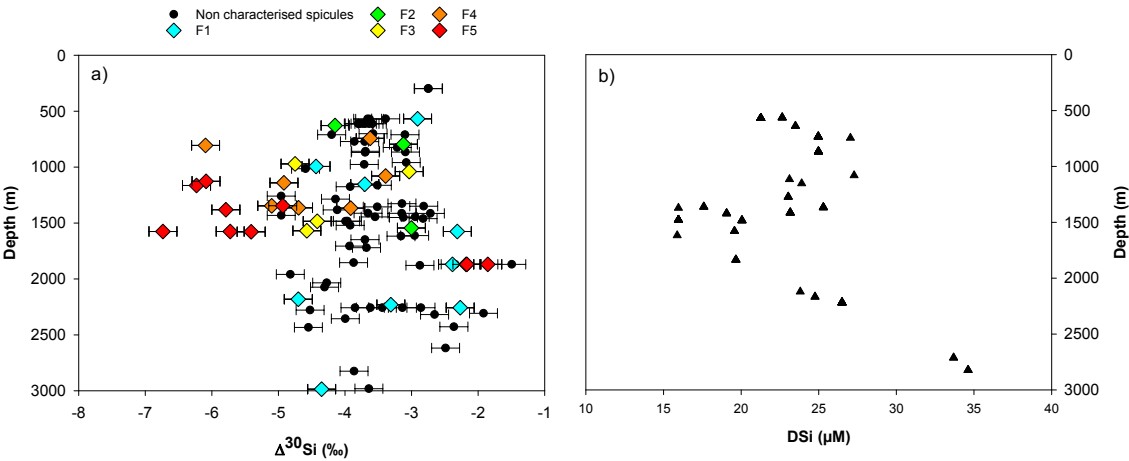

**Figure 7.** a) Apparent Si isotopic fractionation ($\Delta^{30}$Si) by sponges against depth from this study. Coloured diamonds represent the degree of fusion from loose spicules (blue), net-like (green), additional silica coating (yellow), dictyonal framework (orange), dense dictyonal framework (red). b) DSi concentration in the equatorial Atlantic from EBA, EBB, VEM, VAY and GRM.

**Table 1.** Criteria of the five level of sponge spicule fusion. See figure 3 for corresponding picture.

| Fusion level | Given name | Class | Description |
|---|---|---|---|
| F1 | Loose | Demospongiae / Hexactinellida | Single loose microsclere and/or megasclere spicules |
| F2 | Net-like | Hexactinellida | Spicules fused perpendicularly fused by nod forming a relatively regular 2 dimensional net |
| F3 | Parallel coating | Hexactinellida | Spicules fused in parallel and/or multi-angled by additional silica coating |
| F4 | Light dictyonal skeleton | Hexactinellida | Spicules fused/cemented by nod of 6 branches and forming a 3 dimensional framework |
| F5 | Dense dictyonal skeleton | Hexactinellida | Spicules fused/cemented by nod of 6 branches and forming a 3 dimensional framework. Void space between the spicule smaller than F4 and presence of lots of holes within the spicules |

**Table 2.** Summary of Michaelis-Menten enzyme kinetic parameters of sponges used to model $\varepsilon_f$ (figure 6) following equation 2 with $V_{\text{max,p}}$ the the maximum polymerisation rates and $K_{\text{m,P}}$ the half saturation constant of polymerisation.

| Species | $V_{\text{max,p}}$ ($\mu$mol-Si h$^{-1}$ g$^{-1}$) | $K_{\text{m,P}}$ ($\mu$M) | Reference |
|---|---|---|---|
| *Haliclona simulans* | 0.39 | 108.23 | López-Acosta et al. (2018) |
| *Axinella* sp. | 1.74 | 74.478 | Maldonado et al. (2011) |
| *Hymeniacidon perlevis* | 0.127 | 60.441 | López-Acosta et al. (2016) |
| *Suberites ficus* | 0.0148 | 45.92 | López-Acosta et al. (2018) |
| *Halichondria panicea* | 19.33 | 45.438 | Reincke and Barthel (1997) |
| *Tethya citrina* | 0.21 | 29.839 | López-Acosta et al. (2016) |

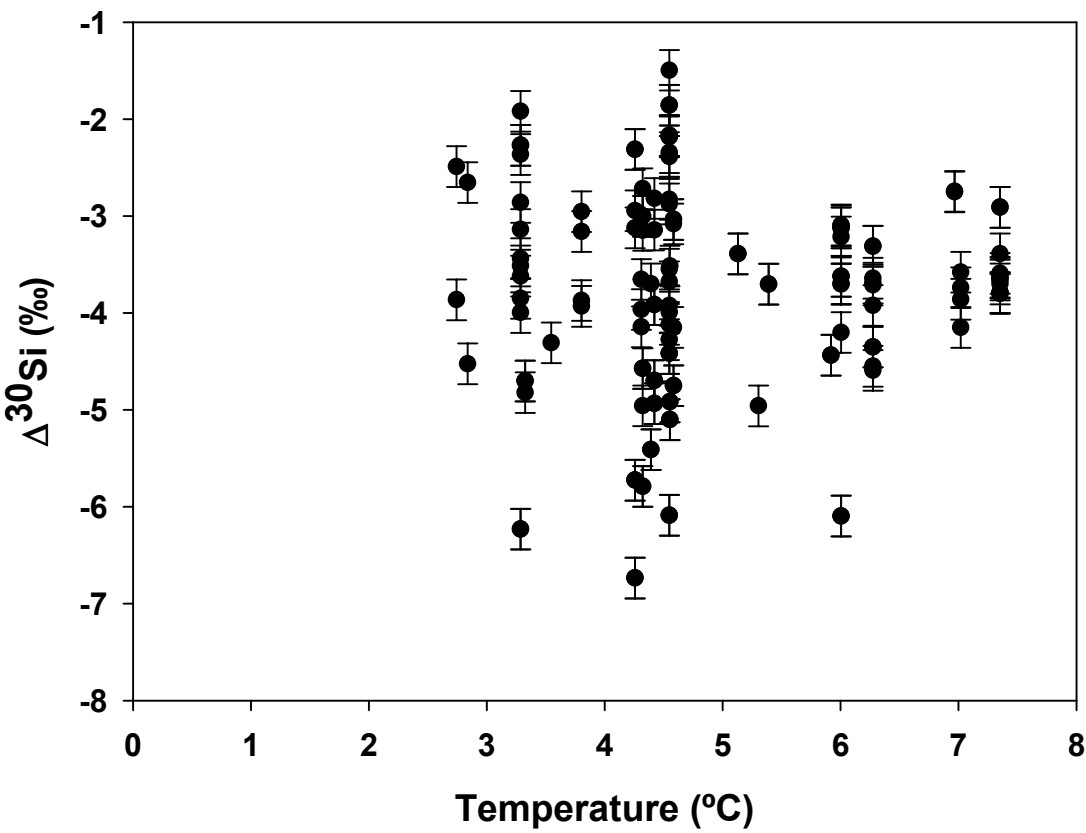

**Figure A1.** Apparent Si isotopic fractionation ($\Delta^{30}$Si) of deep sea sponges from the equatorial Atlantic against the ambient seawater temperature.

**Appendix**

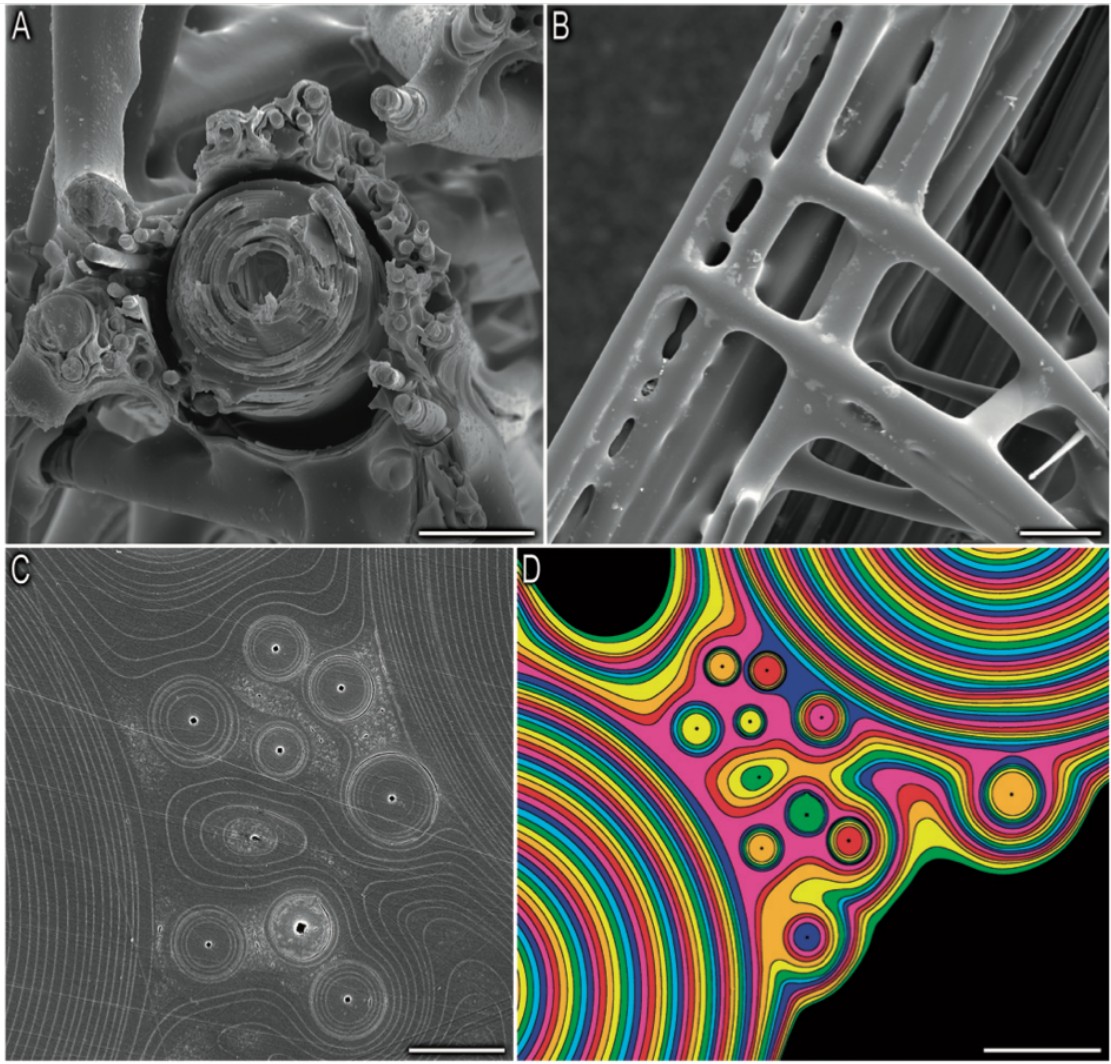

**Figure B1.** Picture from Weaver et al. (2007), Organisational details of the consolidation silica matrix. A) Cross section of the skeletal lattice showing a large spicule surrounded by small spicules, scale bar: 50 $\mu$m. B) External view of the skeletal lattice, scale bar: 100 $\mu$m. C and D) Polished cross-section showing that the cement of the skeletal lattice is made of multiple layers, scale bar: 10 $\mu$m and 20 $\mu$m respectively.

Table A1: Location details and isotopic signature ($\delta^{30}$Si) and apparent Si isotopic fractionation ($\Delta^{30}$Si) of deep sea sponges from the equatorial Atlantic. Taxonomic rank of sample starts with the class (Hexact. for Hexactinellida class and Demosp. for Demospongiae), the order and finally the family. $\delta^{30}$Si$_{spicule}$ and $\delta^{29}$Si$_{spicule}$ are the Si isotopic signatures of the spicules, $\delta^{30}$Si$_{DSi}$ is the Si isotopic signature of seawater, $\Delta^{30}$Si is the apparent Si isotopic fractionation by deep-sea sponges and fusion is the fusion stage defined from SEM images (see table 1 for details. Reproducibility, 2 s.d., is based upon measurements of standards (see main text, section Analytical procedures) and correspond to 0.13 ‰ for $\delta^{30}$Si$_{spicule}$, 0.11 ‰ for $\delta^{30}$Si$_{DSi}$; and 0.17 ‰ for $\Delta^{30}$Si.

| Loc. | Lat (N) | Long (W) | Depth (m) | DSi (μM) | Class. Order. Family | $\delta^{30}$Si$_{spicule}$ (‰) | $\delta^{29}$Si$_{spicule}$ (‰) | $\delta^{30}$Si$_{DSi}$ (‰) | $\Delta^{30}$Si (‰) | Fusion |
|---|---|---|---|---|---|---|---|---|---|---|
| EBA | 9.2358 | 21.5667 | 994 | 27.04 | Hexact. | -3.19 | -1.54 | 1.24 | -4.44 | F1 |
| EBA | 9.4686 | 21.5686 | 1079 | 27.28 | | -2.20 | -1.10 | 1.19 | -3.39 | F4 |
| EBA | 10.1172 | 22.2717 | 298 | 23.5 | Hexact. | -1.36 | -0.74 | 1.39 | -2.75 | |
| EBA | 10.1172 | 22.2717 | 298 | 23.5 | Hexact. | -1.43 | -0.71 | 1.32 | -2.75 | |
| EBA | 9.2064 | 21.2861 | 2073 | 23.81 | Hexact. Lyssacinosida. Euplectellidae | -3.29 | -1.66 | 1.02 | -4.31 | |
| EBA | 10.1517 | 23.6956 | 1413 | 23.16 | Hexact. | -1.97 | -1.01 | 1.18 | -3.15 | |
| EBA | 10.1517 | 23.6956 | 1414 | 23.16 | Hexact. | -1.54 | -0.77 | 1.18 | -2.72 | |
| EBA | 9.7811 | 23.0761 | 1569 | 23.16 | Hexact. Lyssacinosida. Euplectellidae | -3.40 | -1.61 | 1.17 | -4.57 | F3 |
| EBA | 10.1317 | 23.6344 | 1431 | 23.16 | | -3.78 | -1.90 | 1.18 | -4.96 | |
| EBA | 10.2072 | 23.7558 | 1381 | 23.16 | | -4.61 | -2.31 | 1.18 | -5.79 | F5 |
| EBA | 9.6356 | 22.8825 | 1544 | 23.16 | Hexact. | -1.83 | -0.97 | 1.18 | -3.00 | F2 |
| EBA | | | | | Hexact. Sceptrulophora Sceptrulophora incertae sedis | -3.41 | -1.83 | | | F4 |
| EBA | 11.6844 | 22.6069 | 2278 | 33.69 | Hexact. Euplectellidae | -3.32 | -1.66 | 1.20 | -4.52 | |
| EBA | 10.3311 | 23.4453 | 2318 | 33.69 | Hexact. Amphidiscosida Hyalonematidae | -1.45 | -0.72 | 1.20 | -2.65 | |
| EBA | 10.1939 | 23.8964 | 1326 | 25.3 | Hexact. Euplectellidae | -1.97 | -0.98 | 1.18 | -3.14 | |
| EBA | 10.3692 | 24.0494 | 1366 | 25.3 | | -2.74 | -1.39 | 1.18 | -3.91 | F4 |

*Continued on next page*

Table A1 – *Continued from previous page*

| Loc. | Lat (N) | Long (W) | Depth (m) | DSi (μM) | Class. Order. Family | $\delta^{30}Si_{spicule}$ (‰) | $\delta^{29}Si_{spicule}$ (‰) | $\delta^{30}Si_{DSi}$ (‰) | $\Delta^{30}Si$ (‰) | Fusion |
|---|---|---|---|---|---|---|---|---|---|---|
| EBA | 9.2089 | 21.3092 | 1350 | 25.3 | Hexact. | -1.64 | -0.81 | 1.18 | -2.82 | |
| EBA | 9.2089 | 21.3092 | 1364 | 25.3 | | -3.52 | -1.77 | 1.18 | -4.70 | F4 |
| EBA | 9.2092 | 21.3092 | 1345 | 25.3 | | -4.18 | -2.21 | 1.18 | -4.94 | F5 |
| EBB | 5.6133 | 26.9689 | 1575 | 20.05 | Hexact. | -1.09 | -0.59 | 1.23 | -2.31 | F1 |
| EBB | 5.6133 | 26.9689 | 1575 | 20.05 | | -5.51 | -2.71 | 1.23 | -6.74 | F5 |
| EBB | 5.6133 | 26.9689 | 1575 | 20.05 | | -4.50 | -2.14 | 1.23 | -5.73 | F5 |
| EBB | 7.9467 | 28.5336 | 1445 | 20.05 | Demosp. Tetractinellida. *Geodiidae* | -1.72 | -0.83 | 1.23 | -2.95 | |
| EBB | 7.3364 | 28.5236 | 628 | 22.64 | Hexact. Sceptrulophora. *Farreidae* | -2.95 | -1.51 | 1.20 | -4.15 | F2 |
| EBB | 7.2889 | 28.8436 | 701 | 22.64 | Demosp. Tetractinellida. *Vulcanellidae* | -2.38 | -1.23 | 1.20 | -3.58 | |
| EBB | 7.0161 | 28.2778 | 971 | 23.03 | Demosp. *Mycalidae?* | -3.50 | -1.80 | 1.25 | -4.75 | F3 |
| EBB | 6.6592 | 26.9619 | 959 | 23.03 | Hexact. Lyssacinosida. *Euplectellidae* | -1.83 | -0.97 | 1.25 | -3.08 | |
| EBB | 7.3292 | 28.3714 | 611 | 22.64 | | -2.54 | -1.30 | 1.20 | -3.74 | |
| EBB | 7.2764 | 29.3858 | 771 | 22.64 | Hexact. | -2.66 | -1.37 | 1.20 | -3.86 | |
| EBB | 7.5686 | 28.2811 | 2257 | 26.5 | | -2.42 | -1.35 | 1.20 | -3.62 | |
| EBB | 7.5686 | 28.2811 | 2257 | 26.5 | Demosp. | -1.07 | -0.59 | 1.20 | -2.27 | F1 |
| EBB | 5.6833 | 27.0014 | 2824 | 34.62 | Hexact. | -2.77 | -1.45 | 1.09 | -3.86 | |
| EBB | 5.7642 | 27.1419 | 2355 | 26.5 | Demosp. | -2.80 | -1.38 | 1.20 | -4.00 | |
| EBB | 7.5686 | 28.2811 | 2257 | 26.5 | Demosp. | -1.94 | -0.92 | 1.20 | -3.14 | |
| EBB | 7.1094 | 27.5075 | 2618 | 34.62 | Hexact. Amphidis-cosida. *Pheronematidae* | -1.40 | -0.71 | 1.09 | -2.49 | F1 |
| EBB | | | 2427 | 26.5 | Demosp. | -1.17 | -0.60 | 1.20 | -2.36 | |

*Continued on next page*

Table A1 – Continued from previous page

| Loc. | Lat (N) | Long (W) | Depth (m) | DSi (μM) | Class. Order. Family | $\delta^{30}Si_{spicule}$ (‰) | $\delta^{29}Si_{spicule}$ (‰) | $\delta^{30}Si_{DSi}$ (‰) | $\Delta^{30}Si$ (‰) | Fusion |
|---|---|---|---|---|---|---|---|---|---|---|
| EBB | 5.6017 | 26.9678 | 1450 | 20.05 | | -1.90 | -0.91 | 1.22 | -3.12 | |
| EBB | 7.0061 | 29.19 | 1039 | 23.03 | Hexact. Lyssacinosida. *Euplectellidae* | -1.79 | -0.96 | 1.25 | -3.04 | F3 |
| EBB | 7.4347 | 28.5661 | 2307 | 26.5 | | -0.72 | -0.34 | 1.20 | -1.92 | |
| EBB | 5.6214 | 27.1314 | 1164 | 26.5 | Hexact. | -4.98 | -2.51 | 1.25 | -6.23 | F5 |
| EBB | 5.6217 | 26.9647 | 1162 | 26.5 | Demosp. | -2.26 | -1.15 | 1.25 | -3.51 | |
| EBB | 7.5686 | 28.2811 | 2257 | 26.5 | | -2.65 | -1.44 | 1.20 | -3.85 | |
| EBB | 7.5686 | 28.2811 | 2257 | 26.5 | | -2.24 | -1.27 | 1.20 | -3.44 | |
| EBB | 7.5686 | 28.2811 | 2257 | 26.5 | Demosp. | -1.66 | -0.82 | 1.20 | -2.86 | |
| VEM | 12.3936 | 45.8975 | 1355 | 17.59 | Hexact. | -2.25 | -1.21 | 1.27 | -3.52 | |
| VEM | 10.7019 | 44.4172 | 611 | 21.26 | Demosp. | -2.35 | -1.22 | 1.24 | -3.59 | |
| VEM | 10.7019 | 44.4172 | 611 | 21.26 | Demosp. | -2.56 | -1.45 | 1.24 | -3.80 | |
| VEM | 11.8611 | 46.6597 | 1140 | 17.59 | Hexact. Hexasterophora incertae sedis | -3.65 | -1.89 | 1.27 | -4.92 | F4 |
| VEM | 10.8975 | 44.5472 | 1175 | 24.97 | Hexact. Lyssacinosida. *Euplectellidae* | -2.65 | -1.35 | 1.27 | -3.92 | |
| VEM | 11.4853 | 45.0239 | 1648 | 19.57 | Hexact. Lyssacinosida. *Rossellidae* | -2.61 | -1.39 | 1.09 | -3.70 | |
| VEM | 11.1483 | 44.8828 | 1578 | 19.57 | | -4.32 | -2.16 | 1.09 | -5.41 | F5 |
| VEM | | | 1345 | 17.59 | Hexact. | -3.83 | -2.05 | 1.27 | -5.10 | F4 |
| VEM | 11.7417 | 45.4706 | 1014 | 24.97 | Hexact. | -3.32 | -1.73 | 1.27 | -4.59 | |
| VEM | 11.8283 | 46.7042 | 568 | 21.26 | | -2.10 | -1.04 | 1.29 | -3.39 | |
| VEM | 11.8283 | 46.7042 | 568 | 21.26 | Demosp. | -1.62 | -0.91 | 1.29 | -2.91 | F1 |
| VEM | 11.6189 | 45.1725 | 976 | 24.97 | Demosp. Desmacellida | -2.42 | -1.28 | 1.29 | -3.71 | F1 |
| VEM | 10.8033 | 44.6075 | 2230 | 24.97 | Hexact. | -2.37 | -1.23 | 0.94 | -3.31 | F1 |

*Continued on next page*

Table A1 – *Continued from previous page*

| Loc. | Lat (N) | Long (W) | Depth (m) | DSi (μM) | Class. Order. Family | $\delta^{30}Si_{spicule}$ (‰) | $\delta^{29}Si_{spicule}$ (‰) | $\delta^{30}Si_{DSi}$ (‰) | $\Delta^{30}Si$ (‰) | Fusion |
|---|---|---|---|---|---|---|---|---|---|---|
| VEM | 10.7903 | 44.6086 | 2985 | 24.97 | Hexact. Lyssacinosida. *Euplectellidae* | -3.41 | -1.64 | 0.94 | -4.35 | F1 |
| VEM | 10.7019 | 44.4172 | 611 | 21.26 |  | -2.40 | -1.24 | 1.24 | -3.64 | |
| VEM | 10.7019 | 44.4172 | 611 | 21.26 | Demosp. | -2.55 | -1.34 | 1.24 | -3.79 | |
| VEM | 11.8533 | 44.6856 | 2433 | 24.97 | Hexact. | -3.26 | -1.64 | 1.29 | -4.55 | |
| VEM | 10.8181 | 45.3131 | 858 | 24.97 | Demosp. | -2.40 | -1.24 | 1.29 | -3.69 | |
| VEM | 11.0006 | 44.835 | 2981 | 24.97 | Hexact. Lyssacinosida. *Euplectellidae* | -2.70 | -1.43 | 0.94 | -3.64 | |
| VEM | 12.1264 | 44.6475 | 570 | 21.26 |  | -2.34 | -1.22 | 1.29 | -3.63 | |
| VEM | 12.1361 | 44.575 | 569 | 21.26 | Demosp. | -2.37 | -1.25 | 1.29 | -3.66 | |
| VEM | 10.7019 | 44.4172 | 611 | 21.26 | Demosp. | -2.41 | -1.21 | 1.29 | -3.70 | |
| VEM | 10.7019 | 44.4172 | 611 | 21.26 | Demosp. | -2.50 | -1.29 | 1.29 | -3.79 | |
| VAY | 16.8242 | 50.8497 | 1259 | 23.11 | Demosp. | -3.89 | -2.13 | 1.07 | -4.96 | |
| VAY | | | 1285 | 19.06 | Hexact. | -2.92 | -1.55 | 1.22 | -4.14 | |
| VAY | 14.8525 | 48.2594 | 1617 | 19.66 | Demosp. Poeciloscle-rida. *Hymedesmiidae* | -2.02 | -1.03 | 1.14 | -3.16 | |
| VAY | 15.0689 | 48.3697 | 1483 | 19.06 | | -2.82 | -1.47 | 1.14 | -3.96 | |
| VAY | 14.9589 | 48.4347 | 1959 | 24.76 | Hexact. | -3.66 | -1.82 | 1.16 | -4.82 | |
| VAY | 16.1353 | 49.5825 | 1854 | 19.66 | Demosp. Tetractinellida. *Geodiidae* | -2.71 | -1.34 | 1.16 | -3.87 | |
| VAY | 17.3733 | 49.2044 | 1612 | 19.66 | | -1.81 | -0.89 | 1.14 | -2.95 | |
| VAY | 17.0739 | 49.2644 | 1706 | 19.66 | Hexact. Sceptrulophora. *Farreidae* | -2.77 | -1.41 | 1.16 | -3.93 | |
| VAY | 14.8681 | 48.2394 | 1412 | 19.06 | Demosp. | -2.51 | -1.29 | 1.14 | -3.65 | |
| VAY | 14.9889 | 48.1511 | 795 | 24.98 | Hexact. | -1.89 | -0.94 | 1.24 | -3.12 | F2 |

*Continued on next page*

Table A1 – *Continued from previous page*

| Loc. | Lat (N) | Long (W) | Depth (m) | DSi (μM) | Class. Order. Family | $\delta^{30}Si_{spicule}$ (‰) | $\delta^{29}Si_{spicule}$ (‰) | $\delta^{30}Si_{DSi}$ (‰) | $\Delta^{30}Si$ (‰) | Fusion |
|---|---|---|---|---|---|---|---|---|---|---|
| VAY | 14.9914 | 48.1711 | 806 | 24.98 | Hexact. Sceptrulophora. *Sceptrulophora incertae sedis* | -4.86 | -2.43 | 1.24 | -6.10 | F4 |
| VAY | 14.9733 | 48.1772 | 772 | 24.98 | Hexact. | -2.47 | -1.26 | 1.24 | -3.70 | |
| VAY | 15.3689 | 48.4011 | 710 | 24.98 | Hexact. | -1.86 | -0.96 | 1.24 | -3.10 | |
| VAY | 16.0633 | 48.1197 | 1153 | 23.9 | Demosp. Merliida ? | -2.48 | -1.26 | 1.22 | -3.70 | F1 |
| VAY | 16.0633 | 48.2014 | 824 | 24.98 | Hexact. | -1.98 | -1.01 | 1.24 | -3.22 | |
| VAY | 14.8953 | 48.13 | 868 | 24.98 | Hexact. | -2.46 | -1.27 | 1.24 | -3.70 | |
| VAY | 15.3689 | 48.4011 | 710 | 24.98 | Demosp. Desmacellida | -2.97 | -1.40 | 1.24 | -4.20 | |
| VAY | 15.7583 | 48.2117 | 742 | 24.98 | Hexact. | -2.39 | -1.20 | 1.24 | -3.62 | F4 |
| VAY | 16.2986 | 48.1542 | 865 | 24.98 | | -1.86 | -0.90 | 1.24 | -3.09 | |
| VAY | 14.9833 | 50.9286 | 2181 | 24.76 | Hexact. Amphidiscosida. *Pheronematidae* | -3.59 | -1.76 | 1.12 | -4.70 | F1 |
| GRM | 16.0847 | 51.0883 | 1484 | 15.96 | Hexact. | -3.04 | -1.50 | 1.38 | -4.42 | F3 |
| GRM | 17.4306 | 53.1831 | 1869 | 15.96 | Hexact.* | -0.51 | -0.38 | 0.99 | -1.50 | F5 |
| GRM | 17.4306 | 53.1831 | 1869 | 15.96 | Hexact. Tip* | -1.18 | -0.64 | 0.99 | -2.17 | F5 |
| GRM | 17.4306 | 53.1831 | 1869 | 15.96 | Hexact. Base* | -0.87 | -0.52 | 0.99 | -1.86 | F5 |
| GRM | 17.4306 | 53.1831 | 1869 | 15.96 | Demosp. Tetractinellida. *Geodiidae* | -1.20 | -0.64 | 0.99 | -2.18 | F1 |
| GRM | 16.2044 | 51.1544 | 1460 | 15.96 | Demosp. Tetractinellida. *Ancorinidae* | -1.45 | -0.74 | 1.38 | -2.83 | |
| GRM | 15.4167 | 51.0833 | 1520 | 15.96 | Demosp. *Tetractinellida* | -2.55 | -1.29 | 1.38 | -3.92 | |
| GRM | 15.655 | 51.2294 | 2034 | 15.96 | Hexact. Lyssacinosida. *Euplectellidae* | -3.29 | -1.66 | 0.99 | -4.28 | |
| GRM | 17.4281 | 51.0853 | 1878 | 15.96 | Demosp. | -1.89 | -0.90 | 0.99 | -2.88 | |

*Continued on next page*

Table A1 – *Continued from previous page*

| Loc. | Lat (N) | Long (W) | Depth (m) | DSi (μM) | Class. Order. Family | $\delta^{30}Si_{spicule}$ (‰) | $\delta^{29}Si_{spicule}$ (‰) | $\delta^{30}Si_{DSi}$ (‰) | $\Delta^{30}Si$ (‰) | Fusion |
|---|---|---|---|---|---|---|---|---|---|---|
| GRM | 15.4022 | 51.0833 | 1445 | 15.96 | Demosp. | -2.17 | -1.05 | 1.38 | -3.55 | |
| GRM | 15.6253 | 51.1022 | 1127 | 15.96 | Demosp. Biemnida | -4.71 | -2.31 | 1.38 | -6.09 | F5 |
| GRM | 16.6861 | 53.7225 | 1720 | 15.96 | Hexact. | -2.30 | -1.20 | 1.38 | -3.68 | |
| GRM | 15.8794 | 51.3033 | 1382 | 15.96 | Hexact. | -2.74 | -1.40 | 1.38 | -4.12 | |
| GRM | 17.4281 | 53.2044 | 1869 | 15.96 | Demosp. | -1.36 | -0.73 | 0.99 | -2.35 | |
| GRM | 16.0847 | 51.0883 | 1484 | 15.96 | Demosp. | -2.61 | -1.25 | 1.38 | -3.99 | |
| GRM | 17.4306 | 53.1831 | 1869 | 15.96 | Demosp. | -1.40 | -0.80 | 0.99 | -2.39 | F1 |