# Peer review of "SILICON ISOTOPES OF DEEP-SEA SPONGES: NEW INSIGHTS INTO BIOMINERALISATION AND SKELETAL STRUCTURE"

_Biogeosciences, 2018_

## Referee Comment (RC1) · Anonymous Referee #1 · 5 Aug 2018

This manuscript addresses observed differences in the apparent silicon isotope fractionation during spicule formation of marine sponges. The authors compare their new and relatively large dataset from the equatorial Atlantic Ocean to previously published data in an attempt to address the mechanisms influencing the wide range of silicon isotope compositions observed for sponges in the marine environment. The authors present an interesting and plausible argument correlating the apparent silicon isotope fractionation of marine sponges to their skeletal morphology, namely the degree of marine sponge spicule skeletal fusion. The data are interesting and are likely represent a substantial contribution to the scientific community, however the manuscript does have some major shortcomings, as discussed below. Therefore, my overall recommendation

is that this manuscript be accepted for publication after moderate/major revision.

Moderate/Major revisions

The revisions described directly below are for specific sections that require more detail and better structural organization. However, the presentation of the manuscript has several (minor) imprecisions that become very distracting and unfortunately result in an unclear presentation of the scientific approach and the discussion of the data (see Problems with clarity). I have described these 'minor' revisions in a separate section below, but there are several points that need to be addressed to ensure a clear and precise message for the manuscript.

Section 3.2 – Much of this section would be better suited in the methods section, perhaps alongside the SEM work? For example, 'five levels of fusion defined here as F1, F2, F3, F4, and F5 (Table 1). In this section, it is unclear in the text what technique was employed to determine fusion stage. It is clear that SEM imagery was used, but how were the analysed samples chosen? Was it random? Were the analyses performed after the d30Si measurements had been finalized? Please explain.

In addition, the relationship between the fusion stage data and the measured d30Si of the spicules is really quite interesting, and is the basis for a major argument presented in this manuscript, however, this section falls short of describing the results. In particular, the principal 'results' presented in this section are contained in only one sentence (P5 L20-21), a sentence that is difficult to understand. I would suggest the authors revisit this section and provide a better description of how the d30Si of the spicules is strongly associated with different fusion stages.

I also think that figure 4 should be modified slightly. I noticed that the authors had included the data incorporated into making the boxplot for Fusion stage 5, but not the other stages. I actually appreciate the F5 data being presented like this and would prefer if all fusion stages (1-5) were presented in a similar manner.

Section 4.3

P 9 L3-23. – This paragraph is very difficult to follow. I would suggest the following modifications. (1) make a new paragraph starting at line 7 (Figure 6 shows the results. . .) (2) make a new paragraph starting at line 15 (In lopez-Acosta et al. (2016), . . .), (3) change L18-19 to: 'A hypothesis is that. . .' (4) rephrase L18-19 – it is not clear as to why a Km of 10 uM was chosen, nor that the Low Km simulation was included in Figure 6. Also, what was the KMp hat Lopez-Acosta – why did you chose 10 uM?

P9 L26-31 –The argument presented here does not seem plausible or there is something missing in the text. How is efflux [rate] alone influenced by whether a bonding reaction is reversible or not? Can you provide some reasoning here? Even if the bonds are being created and/or destroyed simultaneously, would Si be removed from the organism? Further, the sentence on L29-31 is unclear. The Km of what organism? Or is this theoretical? Most organisms listed in the table have a Km that is much greater than 10 uM (i.e. 29.8-74.5 uM) therefore decreasing the Km to 10 uM doesn't seem likely, or am I missing something? Could you explain what you mean by 'the fractionation due to the efflux..'? Finally, please provide more information regarding the model that was presented as High E efflux and high Km. These are generally not very well described in the text.

Problems with clarity

P1 L7 – what anomalies? Anomalous compared to what? Up until this point no anomalies have been described. Please include a sentence to describe what you mean by anomalies.

P1 L8 – extremely light d30Si signatures? Compared to what? This was not described. Please clarify.

P1 L 10 – please clarify what you mean by spicule types. . .

P1 L 15 – molecular fossil what? The molecular fossil record? Please clarify.

P1 L20 – Do you mean, Of the biomineralizing sponges? Please clarify.

P1/2 L21-2 – This sentence is unclear and I do not quite understand what the authors are trying to convey. This section needs to be developed a bit more and have a stronger link to the previous sentences so that I can understand why the authors wanted to include this information.

P2 L12 – Please provide more detail. Loose where within the skeletal framework? Can the authors please provide a better structural description here? Also what is meant by "…and they have a cellular organization." What kind of organization? Is it unique to each species? Also, the authors need to clarify that this sentence is discussing demosponge mega- and micro-scleres. As it is written, this is not clear in the text.

P2 L15 – what do you mean by rays? Spines along one of the three axes?

P2 L16 – please clarify what you mean by secondary silica.

P2 L17-18 – It is unclear as to why you have included this sentence. Please provide context and improve the conclusion of this paragraph.

P2 L23 – please clarify what you mean by 'sensitive to their environment'. How does growth rate and immobility make sponges sensitive to their environment? This point is unclear.

P2 L 26 – this sentence needs to be reorganized – De La Rocha did not introduce the silicon isotopic composition of biogenic silica.

P3 L17 – the sentence '..by analyzing d30Si along the sponge skeleton?' is not clear, please rephrase.

P3 L18-19 – This sentence is not clear. What precisely is being investigated? Please rephrase.

P3 L22-25 – please rephrase this sentence, it is not clear.

P3 L26-27 – Please provide a list of the samples that were dried, preserved in ethanol and frozen. Please detail where exactly the samples were shipped in the UK.

P3 L28 – was the identification of the specimens to the major sponge classes carried out on subsamples that were preserved or were they fresh?

P3 L29-30 – what is the status of these identifications? Will the species ID be published as an appendix in this paper? Elsewhere? What journal?

P4 L7 – please rephrase the sentence 'If remaining, lithogenic material was removed by hand'. It is unclear.

P4 L7 – Please clarify where the subsample originates. Is it cleaned? Has the lithogenic material been removed? This is not clear.

P4 L13 – Please clarify and rephrase 'Reynolds et al. (2006) modification.'

P5 L7 – How was the D30Si calculated? There are no d30Si data for seawater presented in table A1.

P5 L24-28 – This paragraph could be improved. It is not very descriptive and there is no flow. It reads more like a set of bullet points with, in some cases, poor grammar. Please explain why this information is important, for example, why has 'particular attention been paid to samples with a D30Si larger than 5 permil'? Do all samples show a common feature or just the samples that have a D30Si larger than 5 permil? The information is not abundantly clear from the text and needs to be clarified.

P6 L3 – what studies? Please provide references.

P6 L5 – please define epsilon f. what does it mean?

P6 L26-27 – What do you mean by 'Despite the small range of temperature' – in this dataset? The Hendry and Robinson 2012 dataset? Please clarify.

P6 L29 – please explain what you mean by low concentration? What is the range?

P7 L4-7 – where are these data compiled? Please provide a reference or an appendix. Also, the authors need to be cautious about using the Wille et al. 2010 data set since they did not measure the d30Si of seawater where their collected sponges resided. Their estimates for d30Si came from Cardinal et al. 2005. It would be a good idea to mention this in the text.

Minor revisions

Title: The authors do not provide any new information regarding the influence of biomineralisation on the silicon isotope composition of deep-sea sponges and I would recommend that they change the title to: "Silicon isotopes of deep-sea sponges: New insights from their skeletal structure"

Plurals: please check over text for plural usage. Often, words are incorrectly pluralized. Please correct throughout the manuscript. Here are a few examples from the abstract:

P1 L9 – change 'insights' to 'insight', change 'process' to 'process(es)'

P1 L10 – change 'isotopes' to 'isotope'

Definitions (e.g. Si, DSi, BSi ïĄď30Si, ïĄěf): Please define these abbreviated terms correctly, and once defined, continue to use them instead of their non-abbreviated form. Check throughout the text.

Fractionation: Throughout the text the authors use the word fractionation but often do not describe what is being fractionated (e.g. silicon isotopes). Sometimes the word silicon fractionation is used, when the authors presumably mean silicon isotope fractionation. Please check throughout the text and correct this oversight. I have included a few examples below:

P1 L4 – add 'silicon' to 'apparent isotopic fractionation'

P1 L7 – add 'silicon' to 'isotopic fractionation'

P3 L5 – add 'silicon isotope' to 'apparent fractionation factor'

P3 L7 – add 'silicon isotope' to 'fractionation factor'

P3 L16 – chose to use Si or silicon (see section on definitions below)

P3 L16 – add 'silicon isotope' to 'fractionation'

P3 L18 – add 'silicon' to 'isotopic fractionation'

P5 L8– add 'silicon isotope' to 'fractionation'

P5 L21 – add 'silicon isotope' to 'fractionation'

P6 L5 – add 'apparent silicon isotope' to 'fractionation'

P6 L7 – add 'model' to 'Rayleigh-type fractionation' – also, change Raleigh to Rayleigh.

P6 L9 – add 'isotope' to 'Si fractionation'

P6 L16 – add 'silicon isotope' to 'fractionation'

P6 L20 – add 'silicon isotope' to 'fractionation' (three times)

P6 L20 – add 'isotope' to 'Si fractionation'

P6 L29 – add 'silicon isotope' to 'fractionation'

P7 L20 – add 'isotope' to 'fractionation of Si'

P7 L4, P7 L27, P7 L31, P8 L3, P8 L5, P8 L19, P8 L24, P8 L25, P8 L30, P9 L3, P9 L8, P9 L9, P9 L20, P9 L21, P9 L30, P10 L2, P10 L9, Figure 2 caption, Figure 4 caption, Figure 6 caption, Figure 7 caption, Table A1. . .

P1 L1 – change to "The silicon isotope composition (d30Si) of deep-sea sponges' skeletal elements – spicules – reflect the . . ."

P1 L18 – change to (Strehlow et al., 2010 and references therein)

P2 L5 – Please change sentence to ". . ..spicules through the incorporation and deposition of hydrated amorphous silica (SiO2-nH20), otherwise known as bio-silica."

P2 L10-11 – These two sentences would be better merged.

P2 L16 – change "loose" to "loosely attached"

P2 L19 – please clarify what type of sponges (deep-sea/marine) – I think that you mean 'roused' and not 'aroused'. – also, it is the 'marine' silicon cycle.

P2 L21 – remove 'may be' and replace with 'are'

P2 L 22 – remove 'immobility' and replace with 'inability to move'.

P2 L28 – please write 'approximately'

P2 L29 – you need to include a statement about how the silicon isotope composition is expressed as permil. . . for example: '. . .are reported using delta notation as either d29Si or d30Si using the permil (ØL') scale. . ."

P3 L3 – Please include Wille et al. 2010 and Hendry and Robinson 2012 here along with other references.

P3 L18 – replace 'issues' with 'questions' and 'are going to' with 'will'

P3 L22 – 'a remotely operated vehicle' and 'seawater was sampled using'

P3 L25 – please change to '. . .as smaller individuals encrusted on other organisms. . .'

P4 L3 – remove 'taken and', replace 'in' with 'into'

P4 L5 - replace 'in' with 'into'

P4 L7 - replace 'in' with 'with'

P4 L11 – please state the following sentence earlier in the paragraph: 'The cleaning procedure followed the technique in Hendry et al. (2010) and Hendry and Robinson (2012).

P4 L22 – please add 'at the University of Bristol' 'after Bristol Isotope Group facilities'

P4 L22 – change sentence to: '...were repeated at least twice..' and add the word 'methods' after 'Mg doping'

P5 L 23 – please add 'marine' or 'deep-sea'

P6 L8 – remove 'have'

P6 L9 – change to ' which suggests that silicon isotope fractionation in marine sponges is like to be controlled by a mechanism of Si uptake.'

P6 L24-26 –remove 'concentration, supports Dsi concentration being the main factor controlling silicon isotope fractionation'. There still could be other factors such as pressure, salinity, etc.

P7 L21 – please move this information up to L13.

P7 L29 – remove 'the fact'

P8 L6-7 – 'A spicule is composed of hydrated amorphous silica (SiO2...' was already defined on P2 L5. The purpose of this sentence is unclear, please rephrase.

P8 L7 – remove 'The' as in "The biosilicification'

P8 L 21 – change to 'sponge E. aspergillum is comprised of small spicules that are embedded in a silica matrix surrounding a larger

P8 L26 – change to '...solely a result of the differences in organic composition'

P8 L 30 – change to '...Si isotopes by sponges, epsilon f (see equation 2).

P8 L32 – please provide a reference for your definition of efflux (Milligan? Wille? Other?).

P9 L1-2 – please consider changing to 'values from the aforementioned studies in four different laboratory-based sponge culture experiments (summarized in Table 2).'. Remove 'and with KmP and Vmaxp, the maxium polymersation rates.'

P9 L24 – change to ' Biosilicification in sponges results in the condensation...'. Enzyme should be plural.

Figure 2 caption – please add the abbreviated terms d30Si, D30Si and Si(OH)4 to the figure caption.

Figure 7 – please clarify in the caption that these data are only from the current study.

Table 2 – please define the parameters listed in the table in the table caption. Capitalize the first letter of 'reference'

---

## Short Comment (SC1) · 10 Aug 2018

Thank you for your comments, they have been taken into consideration.

---

## Referee Comment (RC2) · Anonymous Referee #2 · 30 Aug 2018

Review of manuscript bg-2018-328 submitted to Biogeosciences by Cassarino et al: Silicon isotopes of deep-sea sponges: new insights into biomineralisation and skeletal structure

Cassarino et al. present new silicon isotope compositions from sponges recovered from the equatorial Atlantic, and from the water they were growing in. From this, they can derive the silicon isotope fractionation associated with sponge spicule formation. In general, this falls within the same range as previous estimates. The general interpretation of sponge silicon isotope fractionation is that it is related to ambient dissolved silicon concentrations, and thus can be used as a proxy for silicon concentrations in

the ancient oceans, provided the silicon isotope composition of the water is known. However, many of the sponges analysed by Cassarino et al. depart from the published trend (e.g. Hendry and Robinson, 2012), which they show is related to the type of spicule the sponge produces. This adds nuance to our understanding of this developing proxy, and implies more care should be taken in its application to paleorecords. The authors present two hypotheses for why some sponge taxa differ in their silicon isotope fractionation, though cannot conclusively answer why.

In general, the paper is well written, and data seem of high quality, the figures are generally clear – though could be improved, and the references reasonably complete and relevant. Overall, I think this paper falls within the scope of Biogeosciences and is worthy of publication after minor to moderate modifications, which I describe below.

One issue that could be easily improved is the presentation of the data. Currently, it is not possible to recreate the authors analysis because the spicule fusion degree isn't given in the data tables – this could be easily remedied, and it should be also possible to code the symbols in the figures so one can see where the different fusion levels fall.

More generally, because it was not possible to plot the data myself, I became a bit confused at parts, regarding what the takeaway message should be. Some samples fall off all the versions of the fractionation-concentration regression: P5L20 makes clear that this is related to the degree of spicule fusion, which I interpreted from the introduction in general and P6L28 & P7L2 specifically to be related to the taxonomy of the samples, specifically whether they were hexactinellid or demosponges. But then P8L24 and the residual tests says this is not the case. It seems the deviation from the 'average' sponge is the most useful indicator, which is what the residual plots are showing – but in the end it's unclear whether or not this is related to fusion degree or to the taxonomy. My feeling is that if Fig. 4 was altered to show some measure of deviation/residual, rather than absolute values, and the discussion altered to reflect this, things would be clearer. Otherwise surely the default interpretation of Fig. 4 is that different taxa like to live in different parts of the ocean? Similarly, I stuggled to follow the rationale for the

discussion in section 4.3 – it seems trivial if you have 5 (?) tunable parameters, you can make a model produce any magnitude of fractionation. Or have I missed the point here? I would recommend trying to emphasise the key point.

Finally, I wonder if the authors have given any thought to whether this difference in Si-isotope fractionating behaviour between different sponge types is something that could be exploited rather than avoided in a paleoceanographic context?

Minor comments - this is a non-exhaustive list of small typographic, etc. errors and some comments/questions. P1L5: "ranges" P1L15: "fossils" P1L18: "and references therein" P1L20: "Since sponges rely" or "because sponges rely" P2L5: either "from" or "of", not both P2L9: An approximate threshold size for distinguishing between micro- and megascleres would be useful P2L24: A reference to Jochum et al. (2017) might be appropriate here P2L29: Phrasing is unclear, P2 Eqn 1: the permil is not necessary. P3L9: Reference to Fontorbe et al. (2016) and/or Fontorbe et al. (2017) might be appropriate here P3L11 "result" P4L5: "to", not "in" P4L7: How do you know all the lithogenic material was removed? Could it be contaminating the samples? P4L12: "prior to isotopic", and "induced" P4L15: Has it been tested that the yield, including the washing step, is quantitative? P4: Were any seawater standards analysed? (AHOLA, from Grasse et al. (2017)) P4L21: "spectrometry" P5L20: The sentence starting "the d30Si_spicules..." is oddly phrases – it could just say 'd30Si_spicules and apparent fractionation both increase...' P6L5: Not sure it's correct to say epsilon can result only from a biological model – epsilon is simply the permil expression of fractionation factor alpha – see Coplen 2011 "Guidelines and recommended terms for expression of stable- isotope-ratio and gas-ratio measurement results" P6L7: I don't see the relevance of mentioning Rayleigh fractionation here, these are samples from all over the ocean, not a single site evolving through time. P7L1: Or, more generally, that different taxa have different epsilons, Km or Vmax – worth mentioning here? P7L10: "in different ways" P7L18: "from Hendry and Robinson" P7L33: "main" P8L4: "impact the fractionation". Also, a bit of skepticism about the ab initio calculations might be warranted. P8L17/18: Is there an indication of uncertainty on the 'thermal analysis' – are 10% and 15% really different? P8L30 (and elsewhere): 'Sponge fractionation' would more correctly read as 'fractionation of silicon isotopes by sponges' or something similar. P8L31: This sentence is missing a verb. P9L6: "described" P9L24: "enzymes" P9L28: "breaking of bonds"

Figures and Tables Figure 1: I guess the colours represent bathymetry – a color bar would be nice, and perhaps another panel showing a cross-section of dissolved silicon concentrations along the sampling transect.

Figure 2, 5 and 6: It would be convenient to show the different spicule types, perhaps color coded somehow, as in figure 7. I also notice that an outlier from Hendry and Robinson isn't shown – would recommend mentioning this somewhere to be clear.

Figure 4: Why are the individual data points only shown for spicule fusion degree F5? More generally, wouldn't it be more useful to plot the residuals as discussed in the main text? Otherwise all this figure could be telling us is that sponges that produce highly fused spicules prefer to live in the deep sea/high Si concentration waters (and Alvarez et al. (2017) recently showed that different groups do seem to have distinct depth preferences). See also comments above.

Figure 5: Would it make sense to plot the residuals for all data for each possible regression? i.e. then one could see where the new data sits with respect to the previously published data. Also, it would good to see a justification for an expression of this form being chosen rather than e.g. linear fits, power laws, etc.

Figure 7 caption: Atlantic.

Table 2: The recent data from López-Acosta et al. (2018) could be incorporated here.

Figure A1: Could the global data compilation also be presented here? In the caption, 'ambient', not 'ambiant'.

Table A1: For this work to be most useful/reproducible, this table should include the

class of fusion degree each sample has been assigned to. I would have like to plot the data myself but was unable to.

Alvarez, B., Frings, P.J., Clymans, W., Fontorbe, G., Conley, D., 2017. Assessing the potential of sponges (Porifera) as indicators of ocean dissolved Si concentrations. Frontiers in Marine Science 4.

Fontorbe, G., Frings, P.J., Christina, L., Hendry, K.R., Carstensen, J., Conley, D.J., 2017. Enrichment of dissolved silica in the deep equatorial Pacific during the Eocene‐Oligocene. Paleoceanography 32, 848-863.

Fontorbe, G., Frings, P.J., De La Rocha, C.L., Hendry, K.R., Conley, D.J., 2016. A silicon depleted North Atlantic since the Palaeogene: Evidence from sponge and radiolarian silicon isotopes. Earth and Planetary Science Letters 453, 67-77.

Grasse, P., Brzezinski, M.A., Cardinal, D., de Souza, G.F., Andersson, P., Closset, I., Cao, Z., Dai, M., Ehlert, C., Estrade, N., Francois, R., Frank, M., Jiang, G., Jones, J.L., Kooijman, E., Liu, Q., Lu, D., Pahnke, K., Ponzevera, E., Schmitt, M., Sun, X., Sutton, J.N., Thil, F., Weis, D., Wetzel, F., Zhang, A., Zhang, J., Zhang, Z., 2017. GEOTRACES inter-calibration of the stable silicon isotope composition of dissolved silicic acid in seawater. Journal of Analytical Atomic Spectrometry.

Hendry, K.R., Robinson, L.F., 2012. The relationship between silicon isotope fractionation in sponges and silicic acid concentration: Modern and core-top studies of biogenic opal. Geochimica et Cosmochimica Acta 81, 1-12.

Jochum, K., Schuessler, J., Wang, X.H., Stoll, B., Weis, U., Müller, W., Haug, G., Andreae, M., Froelich, P., 2017. Whole‐Ocean Changes in Silica and Ge/Si Ratios During the Last Deglacial Deduced From Long‐Lived Giant Glass Sponges. Geophysical Research Letters 44.

López-Acosta, M., Leynaert, A., Grall, J., Maldonado, M., 2018. Silicon consumption kinetics by marine sponges: An assessment of their role at the ecosystem level. Limnology and Oceanography 0.

---

## Referee Comment (RC3) · Anonymous Referee #3 · 30 Aug 2018

Review of the manuscript "SILICON ISOTOPES OF DEEP-SEA SPONGES: NEW INSIGHTS INTO BIOMINERALISATION AND SKELETAL STRUCTURE " by Lucie Cassarino, Christopher D. Coath, Joana R. Xavier, and Katharine R. Hendry This manuscript present new silicon isotopic data of marine sponges and ambient seawater to evaluate the apparent Si isotopic between both Si reservoirs. The wide range of the apparent Si isotopic of Atlantic sponges grown in relatively low and homogeneous Si concentrations is contrary to previous publications which show that the apparent Si isotopic composition of marine sponges is dependent on ambient dissolved Si concertation of seawater. Based on skeletal morphology, the authors found evidence that sponges which experienced different degree secondary silicification, exhibit extremely

light 30Si signatures. The conclusion is that only certain types of spicules can be used for paleo reconstruction of dissolved silicon in seawater. With this new data this manuscript addresses a relevant scientific question how marine sponges can be used to reconstruct the Si availability in the oceans. I agree with reviewer 1 that a better quantification of the levels of spicule fusion would be desirable. Also an identification of the level of fusion within the table A1 in the appendix would be nice with a possible database of the SEM images for each sample. This enables the reader to get an idea of changing spicule morphology and how it is related to the apparent Si isotopic fractionation. This is one important aspect since from the residual plot in Figure 5d it can be seen that compared to the Demospongiae, Hexactinellida exhibit not only the lightest, but the heaviest D30Si residuals (also seen in Figure 4, although mean and median of all sponges in this group are lighter). Here sponges (Hexactinellida and Demospongiae) from the location GRM 17.4306 53.1831 at 1869m show heavy values. What do spicule morphologies of these samples look like how is their degree of fusion? Sadly I was not able to find the sample with the second highest D30Si (figure 5c at >30uM Si(OH)4. From figure 5d it seems to be a Hexactinellida. Saying this, there is a mismatch between figure 5c and Figure 6 although both figures present all published data and all JC094 data. Also please also compare if all data presented in Figure 5c are also displayed in the residual plot 5d. Some question regarding the dissolved Si concentrations: why are the dissolved Si concentrations for e.g. samples of the GRM location always 15.96 uM although they have a distance of more than 250 km and a depth difference of 400m? Can these sites, close to the mid Atlantic ridge, are affected by dissolved Si supplied by hydrothermal vents? How such a supply, although temporally, might influence the morphology of the spicules? Hydrothermal waters enriched in silica seems to promote the development of sponge growth with have thicker spicules compared to sponges that live in Si poor waters? After excluding such external, environmental parameters section 4.2 and 4.3 discuss the effect of organosilicon complexes on Si isotopic fractionation. Within section 4.2 conclude that the absent "difference in the fractionation between Hexactinellida and Demospongiae

classes despite the difference in their spicule composition, suggesting that the large fractionation in sponges that display a dictyonal framework is not solely a result of the organic composition of the spicules but could be controlled by the enzymes that mediate silica deposition." However only Hexactinellida show a higher degree of fusion (figure 1). So why can the development of a silicon matrix or fusion of Si, possibly with the help of organosilicon complexes, can not be a additional Si deposition method for Hexactinellida? I think that the model calculation in 4.3 is valid since differences in Si uptake parameters are between sponges are obvious and will have influence on the apparent Si isotopic fractionation. But expect one very light data at ~125 uM Si(OH)4 (Hendry & Robinson, 2012) Hexactinellida and Demospongiae from compiled previous publications from different oceans do show a relative small range in residual D30Si. I admire that this manuscript double the amount of available sponge Si isotope data, but if uptake parameters between sponges such a large effect on apparent Si isotope fractionation, why it has not been discovered it in the compiled previous datasets? Does this give indication that environmental parameters at some sample location presented here are different compared to previous publications?

---

## Author Comment (AC1) · 26 Sep 2018

Response to Referee #1

First of all, thank you for the time taking to review the paper. On overall, I agree with the comments you have made, and I have re-worked the manuscript. I hope the corrected version and the detailed respond to your review will make it clearer. Your comments are listed and our response/explanation will be written after it in the following paragraph.

Moderate/Major revisions 1) Section 3.2 – Much of this section would be better suited in the methods section, per- haps alongside the SEM work? For example, 'five levels

of fusion defined here as F1, F2, F3, F4, and F5 (Table 1). In this section, it is unclear in the text what technique was employed to determine fusion stage. It is clear that SEM imagery was used, but how were the analysed samples chosen? Was it random? Were the analyses performed after the d30Si measurements had been finalized? Please explain. In addition, the relationship between the fusion stage data and the measured d30Si of the spicules is really quite interesting, and is the basis for a major argument presented in this manuscript, however, this section falls short of describing the results. In particular, the principal 'results' presented in this section are contained in only one sentence (P5 L20-21), a sentence that is difficult to understand. I would suggest the authors revisit this section and provide a better description of how the d30Si of the spicules is strongly associated with different fusion stages.

2) I also think that figure 4 should be modified slightly. I noticed that the authors had included the data incorporated into making the boxplot for Fusion stage 5, but not the other stages. I actually appreciate the F5 data being presented like this and would prefer if all fusion stages (1-5) were presented in a similar manner. (answer) In the corrected version section 3.2 is left at its original place. We do not think that it has to be in the method section because the degrees of fusion were discovered from the SEM images and so are more suited later in the manuscript. We have changed the first sentence of the section (L20-21 p5) to provide more details about the samples chosen for SEM analysis. Furthermore, the end of the section has been edited (L27-28 p5) to provide a better description of the relationship between d30Sispicule and the degree of fusion. Data were only incorporated into the F5 Box plot (figure 4) to show that the large error was caused by only 2 data points. Changes have been made to figure 4 to include data points of each box plot, and the significant relationship curve has been added to support the hypothesis linking the fractionation factor and the degree of fusion. Also figure 4 is now made of two plots with a) the box plot and b) the residual of the fusion stage against the published calibration to support our assertion that there is a positive relationship between the degree of fusion and the fractionation factor.

3) Section 4.3 P 9 L3-23. – This paragraph is very difficult to follow. I would suggest the following modifications. (1) make a new paragraph starting at line 7 (Figure 6 shows the results. . .) (2) make a new paragraph starting at line 15 (In lopez-Acosta et al. (2016), . . .), (3) change L18-19 to: 'A hypothesis is that. . .' (4) rephrase L18-19 – it is not clear as to why a Km of 10 uM was chosen, nor that the Low Km simulation was included in Figure 6. Also, what was the KMp hat Lopez-Acosta – why did you chose 10 uM? (answer) The entire section 4.3 has been edited to make it easier to follow and also the recent sponge culture study, Lopez-Acosta et a., 2018, has been included. The low Km simulation was already included in figure 6 and was/is referred to as Low Km and the Km value of the T. citrina species (Lopez-Acosta 2016) was/is given in table 2, Km T. citrina = 29.84 $\mu$M. For the low Km simulation, Km = 10 $\mu$M was chosen as a contrast with the published Km value, and because it was the Km value that fitted with the lower $\Delta$30Si limit of unfused sponges.

4) P9 L26-31 –The argument presented here does not seem plausible or there is something missing in the text. How is efflux [rate] alone influenced by whether a bonding reaction is reversible or not? Can you provide some reasoning here? Even if the bonds are being created and/or destroyed simultaneously, would Si be removed from the organism? Further, the sentence on L29-31 is unclear. The Km of what organism? Or is this theoretical? Most organisms listed in the table have a Km that is much greater than 10 uM (i.e. 29.8-74.5 uM) therefore decreasing the Km to 10 uM doesn't seem likely, or am I missing something? Could you explain what you mean by 'the fractionation due to the efflux..'? Finally, please provide more information regarding the model that was presented as High E efflux and high Km. These are generally not very well described in the text. (answer) The end of section 4.3 has been edited in order to present the efflux hypothesis with more details and explanations, now L18-30 p10.

Problems with clarity

5) P1 L7 – what anomalies? Anomalous compared to what? Up until this point no anomalies have been described. Please include a sentence to describe what you

mean by anomalies. (answer) The anomalies represent the fractionation factor that fall outside of the calibration curve. We would like to keep the phrase "anomalies in the isotopic fractionation" as we believe that it concisely describes these data.

6) P1 L8 – extremely light d30Si signatures? Compared to what? This was not described. Please clarify. (answer) "compared with previous studies" L9p1 has been added.

7) P1 L 10 – please clarify what you mean by spicule types. . . (answer) "Spicule types" has been chosen here to describe the spicules shape/fusion without being too technical because we would like to avoid too many taxonomic words in the abstract.

8) P1 L 15 – molecular fossil what? The molecular fossil record? Please clarify. (answer) Done L16p1

9) P1 L20 – Do you mean, Of the biomineralizing sponges? Please clarify. (answer) Done, now L21p1

10) P1/2 L21-2 – This sentence is unclear and I do not quite understand what the authors are trying to convey. This section needs to be developed a bit more and have a stronger link to the previous sentences so that I can understand why the authors wanted to include this information. (answer) This sentence explains that the ratio of calcareous and siliceous sponges in the ocean has been changing with time due to the change in nutrient composition in the ocean. The end of the section has been changed to "but this ratio may have varied in the past due to changes in paleo-ocean chemistry (Montanez, 2002) because sponges rely on the ion chemistry of their surrounding water to build their skeleton" L2-3p2

11) P2 L12 – Please provide more detail. Loose where within the skeletal framework? Can the authors please provide a better structural description here? Also what is meant by ". . .and they have a cellular organization." What kind of organization? Is it unique to each species? Also, the authors need to clarify that this sentence is discussing

demosponge mega- and microscleres. As it is written, this is not clear in the text. (answer) L12-15p2 has been edited: "which compose their skeletal structure. Either Mega- and microscleres are loose, unfused, but joined by spongin (collagen protein) (Uriz, 2006). Demospongiae have a cellular organisation i.e composed of cells that form tissues, which themselves form organs, which form an organism."

12) P2 L15 – what do you mean by rays? Spines along one of the three axes? (answer) Now L18p2 The spicules are described with number of axes in the line before. The rays describe one of the "branches".

13) P2 L16 – please clarify what you mean by secondary silica. (answer) L20p2 "layer and or junction" has been added".

14) P2 L17-18 – It is unclear as to why you have included this sentence. Please provide context and improve the conclusion of this paragraph. (answer) The end of the paragraph has been edited, now L20-22p2 "One key feature that distinguishes between Demospongiae and Hexactinellida class is that Hexactinellida are characterised by a syncytial organisation, i.e. tissue composed of cells without individual plasma membrane (Leys and Lauzon, 1998; Maldonado and Riesgo, 2007)."

15) P2 L23 – please clarify what you mean by 'sensitive to their environment'. How does growth rate and immobility make sponges sensitive to their environment? This point is unclear. (answer) This sentence has been edited, now at L26-28p2 "Because of their relatively low growth rate and their immobility, they are sensitive to change of their environment and because an individual sponge can live decades or centuries (Pansini and Pronzato, 1990; Leys and Lauzon, 1998) they can record information over long time periods (Jochum et al., 2017)."

16) P2 L 26 – this sentence needs to be reorganized – De La Rocha did not introduce the silicon isotopic composition of biogenic silica. (answer) Done, now L30p2 "The silicon isotopic composition of biogenic silica ($\delta$30BSi) has been introduced to study the past nutrient utilisation by 30 De La Rocha et al. (1997) and since has been used

to study the silicon cycle (e.g Hendry et al., 2016; Fontorbe et al., 2017)."

17) P3 L17 – the sentence '..by analysing d30Si along the sponge skeleton?' is not clear, please rephrase. (answer) Done, now L20p3 "Can we trace DSi concentration over time by analysing $\delta$30Si sections of sponge skeleton? "

18) P3 L18-19 – This sentence is not clear. What precisely is being investigated? Please rephrase. (answer) "$\delta$30Si" has been added now L23p3

19) P3 L22-25 – please rephrase this sentence, it is not clear. (answer) Done, now L25-28p3 "Sponge samples were collected by remotely operated vehicle (ROV) and seawaters using Niskin bottles attached to CTD rosette system at five stations, EBA, EBB, VEM, VAY and GRM between 298 m and 2985 m (figure 1) aboard the RRS James Cook on the TROPICS cruise (JC094), a West-East cross section in the equatorial Atlantic between âĹij5âŮęN and âĹij15âŮęN, from the 13th October to the 30th November 2013. "

20) P3 L26-27 – Please provide a list of the samples that were dried, preserved in ethanol and frozen. Please detail where exactly the samples were shipped in the UK. (answer) "preserved in ethanol and frozen" has been deleted to not confuse the reader because none of the samples used in this study were preserved in ethanol or frozen (they were preserved by those means for other studies e.g. taxonomic identifications).

21) P3 L28 – was the identification of the specimens to the major sponge classes carried out on subsamples that were preserved or were they fresh? (answer) The major class identification was done on board (L3p4) based on analysis of the fresh sample.

22) P3 L29-30 – what is the status of these identifications? Will the species ID be published as an appendix in this paper? Elsewhere? What journal? (answer) L5p4 "in a separate paper" is added. The paper has not been submitted yet so we cannot give more details about it.
23) P4 L7 – please rephrase the sentence 'If remaining, lithogenic material was removed by hand'. It is unclear. (answer) The lithogenic material is visible by eye on living sponge spicules, we then remove it by hand if remaining. "before further cleaning steps" has been added L12p4 to precise that the spicule undergoes further cleaning steps.

24) P4 L7 – Please clarify where the subsample originates. Is it cleaned? Has the lithogenic material been removed? This is not clear. (answer) Done, L13p4 "A subsample was taken and weighed before going through a final cleaning step" was added.

25) P4 L13 – Please clarify and rephrase 'Reynolds et al. (2006) modification.' (answer) The MAGIC method is a common method used to analyse d30Si of seawater. The following sentence gives brief details of the method. This section has not been changed as other reviewers did not comment on it.

26) P5 L7 – How was the D30Si calculated? There are no d30Si data for seawater presented in table A1. (answer) In the corrected version d30Si of seawater have been added in table A1. It was not done in the first version because $\triangle$30Si = d30Sispicule – d30Siseawater is describe in the section 1.2 (now L9p3.)

27) P5 L24-28 – This paragraph could be improved. It is not very descriptive and there is no flow. It reads more like a set of bullet points with, in some cases, poor grammar. Please explain why this information is important, for example, why has 'particular attention been paid to samples with a D30Si larger than 5 permil'? Do all samples show a common feature or just the samples that have a D30Si larger than 5 permil? The information is not abundantly clear from the text and needs to be clarified. (answer) This paragraph has been entirely edited, see L3-8p6.

28) P6 L3 – what studies? Please provide references (answer) Done, see L11p6

29) P6 L5 – please define epsilon f. what does it mean? (answer) Sentence changed to "Here $\triangle$30Si is defined by the difference between $\delta$30SiSpicules and $\delta$30SiDSi, which

describes the observed apparent Si isotopic fractionation by sponges whereas $\varepsilon f$ is the result from the biological model from Wille et al. (2010) (equation 2)" see L12-13p6

30) P6 L26-27 – What do you mean by 'Despite the small range of temperature' – in this dataset? The Hendry and Robinson 2012 dataset? Please clarify. (answer) Sentence changed, see L6-7p7 "Despite the small range of seawater temperature in this study, our data show no relationship between $\Delta 30Si$ and temperature (figure A1 in appendix)."

31) P6 L29 – please explain what you mean by low concentration? What is the range? (answer) The range was added, see L9p7

32) P7 L4-7 – where are these data compiled? Please provide a reference or an appendix. Also, the authors need to be cautious about using the Wille et al. 2010 data set since they did not measure the d30Si of seawater where their collected sponges resided. Their estimates for d30Si came from Cardinal et al. 2005. It would be a good idea to mention this in the text. (answer) The data compiled for the residual a) are from the data presented in each paper from Hendry and Robison, 2012; Hendry et al., 2010; Wille et al., 2010. They are the same data presented in figure 2. The data from this study are detailed in table A1 with the fusion degree in the corrected version, which allow the reader to reproduce the residual. Because table A1 is very large, we decide to not add the data from previously published papers as they are already available. Wille et al., 2010 did not measure the d30Si of seawater but due to the conservative nature of d30Si in deep water masses the fractionation factor calculated in Willes et al., 2010 is valid.

Minor revisions

33) Title: The authors do not provide any new information regarding the influence of biomin- eralisation on the silicon isotope composition of deep-sea sponges and I would recommend that they change the title to: "Silicon isotopes of deep-sea sponges: New insights from their skeletal structure" (answer) We would like to keep the original title

**BGD**

because despite that the study did not investigate the direct influence of the biomineralization on the silicon isotope but the relationship between the degree of fusion and the Kmp (half saturation constant of polymerisation) implies that there is.

Plurals: please check over text for plural usage. Often, words are incorrectly pluralized. Please correct throughout the manuscript. Here are a few examples from the abstract:

34) P1 L9 – change 'insights' to 'insight', change 'process' to 'process(es)' (answer) Done, now L9p1

35) P1 L10 – change 'isotopes' to 'isotope' (answer) Done, L10p1

36) Definitions (e.g. Si, DSi, BSi ïA Ìĺd'30Si, ïA ÌĺeĚĞf): Please define these abbreviated terms correctly, and once defined, continue to use them instead of their non-abbreviated form. Check throughout the text. (answer) Done throughout the text.

Fractionation: Throughout the text the authors use the word fractionation but often do not describe what is being fractionated (e.g. silicon isotopes). Sometimes the word silicon fractionation is used, when the authors presumably mean silicon isotope fractionation. Please check throughout the text and correct this oversight. I have included a few examples below: P1 L4 – add 'silicon' to 'apparent isotopic fractionation' P1 L7 – add 'silicon' to 'isotopic fractionation' P3 L5 – add 'silicon isotope' to 'apparent fractionation factor' P3 L7 – add 'silicon isotope' to 'fractionation factor' P3 L16 – chose to use Si or silicon (see section on definitions below) P3 L16 – add 'silicon isotope' to 'fractionation' P3 L18 – add 'silicon' to 'isotopic fractionation' P5 L8– add 'silicon isotope' to 'fractionation' P5 L21 – add 'silicon isotope' to 'fractionation' P6 L5 – add 'apparent silicon isotope' to 'fractionation' P6 L7 – add 'model' to 'Rayleigh-type fractionation' – also, change Raleigh to Rayleigh. P6 L9 – add 'isotope' to 'Si fractionation' P6 L16 – add 'silicon isotope' to 'fractionation' P6 L20 – add 'silicon isotope' to 'fractionation' (three times) P6 L20 – add 'isotope' to 'Si fractionation' P6 L29 – add 'silicon isotope' to 'fractionation' P7 L20 – add 'isotope' to 'fractionation of Si' P7 L4, P7 L27, P7 L31, P8 L3, P8 L5, P8 L19, P8 L24, P8 L25, P8 L30, P9 L3, P9 L8, P9 L9, P9 L20, P9 L21,

P9 L30, P10 L2, P10 L9, Figure 2 caption, Figure 4 caption, Figure 6 caption, Figure 7 caption, Table A1. . .

(answer) Because the paper is focused only on the silicon isotopic fractionation the "silicon isotopic fractionation" term was reduced in the previous version to have lighter sentence to read. This has been changed in the corrected version.

37) P1 L1 – change to "The silicon isotope composition (d30Si) of deep-sea sponges' skeletal elements – spicules – reflect the . . ." (answer)Done, L1p1

38) P1 L18 – change to (Strehlow et al., 2010 and references therein) (answer) Done, L19p1

39) P2 L5 – Please change sentence to ". . ..spicules through the incorporation and deposition of hydrated amorphous silica (SiO2-nH20), otherwise known as bio-silica." (answer) Sentence changed to "produce their spicules made of bio-silica (amorphous silica)" L4-5p2

40) P2 L10-11 – These two sentences would be better merged. (answer) New paragraph started L11p2 and the sentences have been merged.

41) P2 L16 – change "loose" to "loosely attached" (answer) We decided to keep "loose" to distinguish between cases where the spicules are attached to each other with secondary silica

42) P2 L19 – please clarify what type of sponges (deep-sea/marine) – I think that you mean 'roused' and not 'aroused'. – also, it is the 'marine' silicon cycle. (answer) Done, now L23p2

48) P2 L21 – remove 'may be' and replace with 'are. (answer) Now L25p2 has not been changed because this subject has been solely highlighted in one paper and the study used one area rather than a compilation to extend their standing Si stock to a global scale.

49) P2 L 22 – remove 'immobility' and replace with 'inability to move'. (answer) Done, now L26/27p2

50) P2 L28 – please write 'approximately'. (answer) Done, now L32p2

51) P2 L29 – you need to include a statement about how the silicon isotope composition is expressed as permil. . . for example: '. . .are reported using delta notation as either d29Si or d30Si using the permil (ØL') scale. . .". (answer) This has been changed to : "Silicon isotopic abundances in samples (SMP) are expressed as $\delta$29Si or $\delta$30Si with the abundance ratio, 29Si/28Si or 30Si/28Si respectively and measured relative to the reference standard (NBS28). The results presented in this study are expressed as permil to be consistent with the International Union of Pure and Applied Chemistry (IUPAC) nomenclature. i.e." (L33p2 to 2p3).

52) P3 L3 – Please include Wille et al. 2010 and Hendry and Robinson 2012 here along with other references. (answer) Done, now L6p3

53) P3 L18 – replace 'issues' with 'questions' and 'are going to' with 'will' (answer) Done, now L21p3

54) P3 L22 – 'a remotely operated vehicle' and 'seawater was sampled using' (answer) Done, now L25p3

55) P3 L25 – please change to '. . .as smaller individuals encrusted on other organisms. . .' (answer) Done, now L1p4

56) P4 L3 – remove 'taken and', replace 'in' with 'into' (answer)Done, now L8p4

57) P4 L5 - replace 'in' with 'into' (answer) Done, now L10p4

58) P4 L7 - replace 'in' with 'with' (answer) Done, now L12p4

59) P4 L11 – please state the following sentence earlier in the paragraph: 'The cleaning procedure followed the technique in Hendry et al. (2010) and Hendry and Robinson (2012). (answer) We decided to keep the cleaning procedure at the end of the paragraph. Now, L16p4

60) P4 L22 – please add 'at the University of Bristol' 'after Bristol Isotope Group facilities' (answer) Done, L28p4

61) P4 L22 – change sentence to: '. . .were repeated at least twice.' and add the word 'methods' after 'Mg doping' (answer) Done, L28p4

62) P5 L 23 – please add 'marine' or 'deep-sea' P6 L 8 – remove 'have' (answer) deep-sea has been added, now L2p6

63) P6 L9 – change to ' which suggests that silicon isotope fractionation in marine sponges is like to be controlled by a mechanism of Si uptake.' (answer) "marine sponges" has been added but we kept "mainly". Now L17p6

64) P6 L24-26 –remove 'concentration, supports Dsi concentration being the main factor controlling silicon isotope fractionation'. There still could be other factors such as pressure, salinity, etc. (answer) This sentence has been kept as "which supports DSi concentration being the main factor controlling Si isotopic fractionation" (L6p7).

65) P7 L21 – please move this information up to L13. (answer) Done, now L25p7

66) P7 L29 – remove 'the fact' (answer) Done, now L9p8

67) P8 L6-7 – 'A spicule is composed of hydrated amorphous silica ($SiO_2$. . .' was already defined on P2 L5. The purpose of this sentence is unclear, please rephrase. (answer) This sentence introduces the reader to this new section that focus on the spicule composition, so this information is a key point. This sentence is a reminder like that reader does not need to return to the beginning of the paper to understand the rest of the paragraph. "and organic molecules (Uriz et al., 2003)" has been added at the end of the sentence. Now L21p8

68) P8 L7 – remove 'The' as in "The biosilicification" (answer) Done, now L23p8

69) P8 L 21 – change to 'sponge E. aspergillum is comprised of small spicules that are

embedded in a silica matrix surrounding a larger (answer) Done, now L5p9

70) P8 L26 – change to '. . .solely a result of the differences in organic composition' (answer) Done, now L9-10p9

71) P8 L 30 – change to '. . .Si isotopes by sponges, epsilon f (see equation 2). (answer) Done, now L13p9

72) P8 L32 – please provide a reference for your definition of efflux (Milligan? Wille? Other?). (answer) "(Wille et al., 2010, and references therein)" has been added, now L16p9

73) P9 L1-2 – please consider changing to 'values from the aforementioned studies in four different laboratory-based sponge culture experiments (summarized in Table 2).'. Remove 'and with KmP and Vmaxp, the maxium polymersation rates.' (answer) The end of the paragraph has been changed to "To date, only Reincke and Barthel (1997); Maldonado et al. (2011); López-Acosta et al. (2016) and López-Acosta et al. (2018) have cultured sponges to investigate the 15 Michaelis-Menten enzyme kinetics of sponges. In this section, $\varepsilon$f has been modelled using Km,P and Vmax,p values from these four sponge culture experiments and are summarised in table 2." L16-19p9

74) P9 L24 – change to ' Biosilicification in sponges results in the condensation. . .'. Enzyme should be plural. (answer) Done, now L18p10

75) Figure 2 caption – please add the abbreviated terms d30Si, D30Si and Si(OH)4 to the figure caption. (answer)Done, "a) Silicon isotopic composition of the spicules ($\delta$30SiSpicules) and b) deep sea sponges apparent Si isotopic fractionation ($\Delta$30Si) against DSi" has been added.

76) Figure 7 – please clarify in the caption that these data are only from the current study. (answer) Done, "from this study" has been added.

77) Table 2 – please define the parameters listed in the table in the table caption. Capitalize the first letter of 'reference' (answer) Done, "with Vmax,p the the maximum

polymerisation rates and Km,P the half saturation constant of polymerisation." Has been added.

---

## Author Comment (AC2) · 26 Sep 2018

Response to Referee #2

First of all, thank you for the time taking to review the paper. On overall, I agree with the comments you have made, and I have re-worked the manuscript. I hope the corrected version and the detailed respond to your review will make it clearer. Your comments are listed, and our response/explanation will be written after it in the following paragraph.

1) One issue that could be easily improved is the presentation of the data. Currently, it is not possible to recreate the authors analysis because the spicule fusion degree isn't

given in the data tables – this could be easily remedied, and it should be also possible to code the symbols in the figures so one can see where the different fusion levels fall. (answer) The data table A1 has been edited and the fusion degree of the spicules is now available. We have change figure 2 , which now has the fusion degree included, but we did not do this for figure 5 and 6 because figure 5 focuses on the comparison between Hexactinellids and Demosponges, and figure 6 is already very colourful due to the 8 $\varepsilon$f simulations. Adding additional colour-coding for the degree of fusion would make both figures unreadable.

2) More generally, because it was not possible to plot the data myself, I became a bit confused at parts, regarding what the takeaway message should be. Some samples fall off all the versions of the fractionation concentration regression: P5L20 makes clear that this is related to the degree of spicule fusion, which I interpreted from the introduction in general and P6L28 & P7L2 specifically to be related to the taxonomy of the samples, specifically whether they were hexactinellid or demosponges. But then P8L24 and the residual tests says this is not the case. It seems the deviation from the 'average' sponge is the most useful indicator, which is what the residual plots are showing – but in the end it's unclear whether or not this is related to fusion degree or to the taxonomy. (answer) Previously L28p6 "a group of hexactinellid sponge" has been changed to "a group of sponge" now L8p7 to avoid confusion.

3) My feeling is that if Fig. 4 was altered to show some measure of deviation/residual, rather than absolute values, and the discussion altered to reflect this, things would be clearer. Otherwise surely the default interpretation of Fig. 4 is that different taxa like to live in different parts of the ocean? (answer) Figure 4 was only representing the samples identified from this study, which come from the deep-water masses of the equatorial Atlantic. This information has been added in the caption to avoid confusion. Furthermore, in the corrected version, figure 4b is added and shows the fractionation factor residual of the 5 degrees of fusion compared to the published calibration curve.

4) Similarly, I stuggled to follow the rationale for the discussion in section 4.3 – it seems

trivial if you have 5 (?) tunable parameters, you can make a model produce any magnitude of fractionation. Or have I missed the point here? I would recommend trying to emphasise the key point. (answer) Section 4.3 has been entirely edited and is now organised in 2 parts: first the Kmp is investigated and then the efflux. Kmp is first investigated due to the relationship between $\varepsilon$f simulations results and the Michaelis-Menten parameters and then pushed to extreme value, Kmp=10$\mu$M, to see the effect on $\varepsilon$f. Because $\varepsilon$f of the dictional skeleton cannot be simulated by Kmp we then investigate the effect of the fractionation of the efflux. The section is now better structured to help the reader.

5) Finally, I wonder if the authors have given any thought to whether this difference in Si- isotope fractionating behaviour between different sponge types is something that could be exploited rather than avoided in a paleoceanographic context? (answer) At the end of section 4.3 we suggest that Si isotopes could potentially be used to investigate cellular uptake and silicification processes, see L29-30p10.

Minor comments - this is a non-exhaustive list of small typographic, etc. errors and some comments/questions.

6) P1L5: "ranges" (answer) Done, now L5p1

7) P1L15: "fossils" (answer) Now "fossil records" L16p1

8) P1L18: "and references therein" (answer)Done, now L19p1

9) P1L20: "Since sponges rely" or "because sponges rely" (answer)Done, now L2p2

10) P2L5: either "from" or "of", not both (answer) Done, change to ''made of", now L5p2

11) P2L9: An approximate threshold size for distinguishing between micro- and megascleres would be useful. (answer) Size has been added "megascleres (up to and beyond 300 $\mu$m) and microscleres (up to 50 $\mu$m)", now L9p2

12) A reference to Jochum et al. (2017) might be appropriate here P2L29: (answer) Done, now L28p2

13) Phrasing is unclear, P2 Eqn 1: the permil is not necessary. (answer) We haven't changed it because x1000 is not included into the equation the permil symbol show that the delta value is express in permil unit.

14) P3L9: Reference to Fontorbe et al. (2016) and/or Fontorbe et al. (2017) might be appropriate here (answer) Reference Fontorbe et al., 2017 has been added, now L13p3

15) P3L11 "result" (answer) Done, now L14p3

16) P4L5: "to", not "in" (answer) change ''to'' by ''into'', now L10p4

17) P4L7: How do you know all the lithogenic material was removed? Could it be contaminating the samples? (answer) The sponge spicules analysed here are from live sponges. The potential lithogenic material is visible by eyes if remaining after the first steps of cleaning, and it is possible to remove it. Furthermore, further cleaning steps are done with high concentrated acid, which results in very clean spicules, so we are confident that the spicules are free of lithogenics. The sentence has been edited to "If remaining, lithogenic material was removed by hand before further cleaning steps. A subsample was taken and weighed before going through a final cleaning step", see L12-13p4

18) P4L12: "prior to isotopic", and "induced" (answer) Done, L18p4

19) P4L15: Has it been tested that the yield, including the washing step, is quantitative? (answer) Yes, the yield was tested and is now included in the method, see L23p4 "The yield recovery of Si is equivalent to 92.1%"

20) P4: Were any seawater standards analysed? (AHOLA, from Grasse et al. (2017)). (answer) We have analysed the ALOHA Deep water standard using this method and have added the results, see L1-3p5 '' The new seawater standard ALOHA deep was

analysed as an additional quality check, and yielded values within error of those obtained during an interlaboratory (Grasse et al., 2017): Aloha deep: $\delta 30Si = 1.08\pm 0.12$ ‰ and $\delta 29Si = 0.58\pm 0.12$ ‰ (2 s.d, n = 4)."

21) P4L21: "spectrometry" (answer) Done, Now L27p4

22) P5L20: The sentence starting "the d30Si_spicules. . ." is oddly phrases – it could just say 'd30Si_spicules and apparent fractionation both increase. . .' (answer) The end of the paragraph has been changed to '' It is observed that $\delta 30SiSpicules$ and $\Delta 30Si$ show an enrichment in light isotopes with higher degree of spicule fusion (see figure 4)." Now L27-28p5

23) P6L5: Not sure it's correct to say epsilon can result only from a biological model – epsilon is simply the permil expression of fractionation factor alpha – see Coplen 2011 "Guidelines and recommended terms for expression of stable- isotope-ratio and gas-ratio measurement results" (answer) We have changed the sentence to "Here $\Delta 30Si$ is defined by the difference between $\delta 30SiSpicules$ and $\delta 30SiDSi$, which describes the observed apparent Si isotopic fractionation by sponges whereas $\varepsilon f$ is the result from the biological model from Wille et al. (2010) (equation 2)" (L12-14p6) to distinguish the observe and the calculated fractionation factor, which follow the previous published paper.

24) P6L7: I don't see the relevance of mentioning Rayleigh fractionation here, these are samples from all over the ocean, not a single site evolving through time. (answer) This sentence is to emphasize that sponge Si isotopic fractionation does not follow the Raleigh type fractionation observed in diatoms.

25) P7L1: Or, more generally, that different taxa have different epsilons, Km or Vmax – worth mentioning here? (answer) The end of the paragraph has been changed but not the last sentences.

26) P7L10: "in different ways" (answer) Done, now L22p7

27) P7L18: "from Hendry and Robinson" (answer) Done, now L30p7

28) P7L33: "main" (answer) Done, L15p8

29) P8L4: "impact the fractionation". (answer) Done, L19p8 Also, a bit of skepticism about the ab initio calculations might be warranted.

30) P8L17/18: Is there an indication of uncertainty on the 'thermal analysis' – are 10% and 15% really different? (answer) The uncertainties on those values are not known, to our knowledge but to support it another experiment and reference is added, see L33p8 L2p9 "Furthermore, SDS-PAGE analysis of Hexactinellid Euplectella aspergilum has shown that the proteins of its axial filament have higher molecular weights than those isolated in demosponges (Weaver and Morse, 2003),"

31) P8L30 (and elsewhere): 'Sponge fractionation' would more correctly read as 'fractionation of silicon isotopes by sponges' or something similar. (answer) Done

32) P8L31: This sentence is missing a verb. (answer) The sentence has been changed to '' The Si isotopic fractionation by sponges is assumed to occur during Si uptake and during internal spicule formation with spicule formation being a function of Si influx and efflux from the sclerocyte (Milligan et al., 2004)", see L13-14p9

33) P9L6: "described" (answer) Done, now L24p9

34) P9L24: "enzymes" (answer) Done, now L23p10

35) P9L28: "breaking of bonds" (answer) Done, now L22p10

36) Figures and Tables Figure 1: I guess the colours represent bathymetry – a color bar would be nice, and perhaps another panel showing a cross-section of dissolved silicon concentrations along the sampling transect. (answer) Colour bar for the bathymetry and DSi cross section have been added.

37) Figure 2, 5 and 6: It would be convenient to show the different spicule types, perhaps color coded somehow, as in figure 7. (answer) The colour coding is mentioned

and commented in general comments. I also notice that an outlier from Hendry and Robinson isn't shown – would recommend mentioning this somewhere to be clear. (answer) In fact, the core top samples from Hendry and Robinson (2012) are not included. This is now mentioned in figure 2 caption.

38) Figure 4: Why are the individual data points only shown for spicule fusion degree F5? (answer) Data points for all the fusion degree have been added. In the first version, only F5 was detailed to show why the error bars was so large. More generally, wouldn't it be more useful to plot the residuals as discussed in the main text? (answer) A residual test is now added, see figure 4b. Otherwise all this figure could be telling us is that sponges that produce highly fused spicules prefer to live in the deep sea/high Si concentration waters (and Alvarez et al. (2017) recently showed that different groups do seem to have distinct depth preferences). See also comments above. (answer) Figure 4a only show the samples with identified fusion degree from the deep water of the equatorial Atlantic indicating that there is a significant relationship between the Si isotopic fractionation and the degree of fusion. We have added the following to the caption of figure 4: '' Data plotted here correspond to the samples from the equatorial Atlantic with identified fusion stage i.e. coloured diamond of b) and so occupy a very narrow range of DSi. b) $\Delta30Si$ residual of samples with identified fusion stage compared to the published calibration (best fit 1)."

38) Figure 5: Would it make sense to plot the residuals for all data for each possible regression? i.e. then one could see where the new data sits with respect to the previously published data. (answer) Figure 4b has been added and show the residual test of the fusion degree compared to the previous published data. Also, it would good to see a justification for an expression of this form being chosen rather than e.g. linear fits, power laws, etc. (answer) The best fit equations are following the fractionation factor equation (equation 2), according to published methodology (e.g. Wille et al., 2010; Hendry & Robinson, 2012).

39) Figure 7 caption: Atlantic. (answer) Done

40) Table 2: The recent data from López-Acosta et al. (2018) could be incorporated here. (answer) The recent data from Lopez-Acosta et al., 2018 are now included.

41) Figure A1: Could the global data compilation also be presented here? The temperature data of all data compilation is not available, the caption, 'ambient', not 'ambiant'. (answer) Done

42) Table A1: For this work to be most useful/reproducible, this table should include the class of fusion degree each sample has been assigned to. I would have like to plot the data myself but was unable to. (answer) The fusion degree are now included into table A1.

---

## Author Comment (AC3) · 26 Sep 2018

Response to referee #3

First of all, thank you for the time taking to review the paper. On overall, I agree with the comments you have made, and I have re-worked the manuscript. I hope the corrected version and the detailed respond to your review will make it clearer. Your comments are listed, and our response/explanation will be written after it in the following paragraph.

1) I agree with reviewer 1 that a better quantification of the levels of spicule fusion would be desirable. Also an identification of the level of fusion within the table A1

in the appendix would be nice with a possible database of the SEM images for each sample. This enables the reader to get an idea of changing spicule morphology and how it is related to the apparent Si isotopic fractionation. (answer) The degree of fusion has been added into table A1. However, we did not add pictures because the table is already very large, and examples of the various degrees of fusion are given in figure 3. It is mentioned in the sample availability section that images are available upon request. This is one important aspect since from the residual plot in Figure 5d it can be seen that compared to the Demospongiae, Hexactinellida exhibit not only the lightest, but the heaviest D30Si residuals (also seen in Figure 4, although mean and median of all sponges in this group are lighter).

2) Here sponges (Hexactinellida and Demospongiae) from the location GRM 17.4306 53.1831 at 1869m show heavy values. What do spicule morphologies of these samples look like how is their degree of fusion? (answer) The fusion degree of the samples from GRM with heavy d30Sispicule are now documented in table A1 and show a F1 and F5 degree of fusion.

3) Sadly I was not able to find the sample with the second highest D30Si (figure 5c at >30uM Si(OH)4. From figure 5d it seems to be a Hexactinellida. Saying this, there is a mismatch between figure 5c and Figure 6 although both figures present all published data and all JC094 data. Also please also compare if all data presented in Figure 5c are also displayed in the residual plot 5d. (answer) There is a mismatch between figure 2, 5c and 6 because not all samples have been identified. We have clarified this point in figure 5 caption to avoid confusion.

4) Some question regarding the dissolved Si concentrations: why are the dissolved Si concentrations for e.g. samples of the GRM location always 15.96 uM although they have a distance of more than 250 km and a depth difference of 400m? (answer) Unfortunately, it was not always possible to collect co-located sponge and water samples: the water sample closed to the sponge location was analysed.

5) Can these sites, close to the mid Atlantic ridge, are affected by dissolved Si supplied by hydrothermal vents? How such a supply, although temporally, might influence the morphology of the spicules? Hydrothermal waters enriched in silica seems to promote the development of sponge growth with have thicker spicules compared to sponges that live in Si poor waters? (answer) We have not discussed the potential effect of hydrothermal supply. Sampling of waters directly above the TAG hydrothermal site did not reveal any anomalies in dissolved Si concentrations and only a very local anomaly in silicon isotopic composition of seawater (Brzezinski & Jones, 2015; Deep Sea Research Part II: Topical Studies in Oceanography volume 116); we did not sample any sponges or waters sufficiently close to any active sites to their to have been an impact on sponge growth or isotopic composition.

6) After excluding such external, environmental parameters section 4.2 and 4.3 discuss the effect of organosilicon complexes on Si isotopic fractionation. Within section 4.2 conclude that the absent "difference in the fractionation between Hexactinellida and Demospongiae classes despite the difference in their spicule composition, suggesting that the large fractionation in sponges that display a dictyonal framework is not solely a result of the organic composition of the spicules but could be controlled by the enzymes that me- diate silica deposition." However only Hexactinellida show a higher degree of fusion (figure 1). So why can the development of a silicon matrix or fusion of Si, possibly with the help of organosilicon complexes, can not be a additional Si deposition method for Hexactinellida? (answer) The end of section 4.2 has been changed to '' This suggests that the large $\Delta 30Si$ in sponges that display a dictyonal framework is not solely a result of the differences in organic composition of the spicules but could also be controlled by the enzymes that mediate silica deposition."

7) I think that the model calculation in 4.3 is valid since differences in Si uptake parameters are between sponges are obvious and will have influence on the apparent Si isotopic fractionation. But expect one very light data at âĹij125 uM Si(OH)4 (Hendry & Robinson, 2012) Hexactinellida and Demospongiae from compiled previous publica-
tions from different oceans do show a relative small range in residual D30Si. I admire that this manuscript double the amount of available sponge Si isotope data, but if uptake parameters between sponges such a large effect on apparent Si isotope fractionation, why it has not been discovered it in the compiled previous datasets? Does this give indication that environmental parameters at some sample location presented here are different compared to previous publications? (answer) An entire section has been added "4.4 Comparison with previous studies The new data set of this study show a wide range of $\Delta$30Si for a small range of DSi concentration compared to previous studies (figure 2) (Hendry and Robinson, 2012; Wille et al., 2010; Hendry et al., 2010). Here spicule shape, in particular the fusion stage, and $\Delta$30Si have been investigated and are closely related, where high fusion stages show very large $\Delta$30Si, which deviate from the existing calibration. Why has this relationship between spicule fusion and $\Delta$30Si not been observed in previous studies? This new data set is composed of 103 samples in which 15 are deviating from the calibration and display a dyctional skeleton. Previous studies are based on fewer samples and all the hexactinellid specimens have been found, except for one, in high DSi environments (higher than 45 $\mu$M) (Hendry and Robinson, 2012; Wille et al., 2010; Hendry et al., 2010). As the spicules in Hexactinellida class can be loose, partially or totally fused, or even cemented by secondary silica (Uriz et al., 2003), it is likely that previous studies only analysed samples with loose spicules (equivalent to F1 here). Furthermore, Hendry and Robinson (2012) piblished one sample with $\Delta$30Si = -6.52 ‰ for DSi = 125 $\mu$M (figure 2). This sample also displayed a fused skeleton but at this date the large fractionation was attributed to the lack of constraint on ambient seawater $\delta$30Si ."

---

## Author Response (AR1)

Response to Referee #1:

First of all, we would like to thank all referees for the time taking to review the paper. On overall, we agree with the comments made, and we have re-worked the manuscript. We hope the corrected version and the detailed respond to the review will make it clearer.
Your comments are in bold and blue and our response/explanation in black.

**Moderate/Major revisions**

**Section 3.2 – Much of this section would be better suited in the methods section, perhaps alongside the SEM work? For example, 'five levels of fusion defined here as F1, F2, F3, F4, and F5 (Table 1). In this section, it is unclear in the text what technique was employed to determine fusion stage. It is clear that SEM imagery was used, but how were the analysed samples chosen? Was it random? Were the analyses performed after the d30Si measurements had been finalized? Please explain.**

**In addition, the relationship between the fusion stage data and the measured d30Si of the spicules is really quite interesting, and is the basis for a major argument presented in this manuscript, however, this section falls short of describing the results. In particular, the principal 'results' presented in this section are contained in only one sentence (P5 L20-21), a sentence that is difficult to understand. I would suggest the authors revisit this section and provide a better description of how the d30Si of the spicules is strongly associated with different fusion stages.**

**I also think that figure 4 should be modified slightly. I noticed that the authors had included the data incorporated into making the boxplot for Fusion stage 5, but not the other stages. I actually appreciate the F5 data being presented like this and would prefer if all fusion stages (1-5) were presented in a similar manner.** In the corrected version section 3.2 is left at its original place. We do not think that it has to be in the method section because the degrees of fusion were discovered from the SEM images and so are more suited later in the manuscript. We have changed the first sentence of the section (L20-21 p5) to provide more details about the samples chosen for SEM analysis. Furthermore, the end of the section has been edited (L27-28 p5) to provide a better description of the relationship between d30Si$_{spicule}$ and the degree of fusion.

Data were only incorporated into the F5 Box plot (figure 4) to show that the large error was caused by only 2 data points. Changes have been made to figure 4 to include data points of each box plot, and the significant relationship curve has been added to support the hypothesis linking the fractionation factor and the degree of fusion. Also figure 4 is now made of two plots with a) the box plot and b) the residual of the fusion stage against the published calibration to support our assertion that there is a positive relationship between the degree of fusion and the fractionation factor.

**Section 4.3 P 9 L3-23. – This paragraph is very difficult to follow. I would suggest the following modifications. (1) make a new paragraph starting at line 7 (Figure 6 shows the results. . .) (2) make a new paragraph starting at line 15 (In lopez-Acosta et al. (2016), . . .), (3) change L18-19 to: 'A hypothesis is that. . .' (4) rephrase L18-19 – it is not clear as to why a Km of 10 uM was chosen, nor that the Low Km simulation was included in Figure 6. Also, what was the KMp hat Lopez-Acosta – why did you chose 10 uM?** The entire section 4.3 has been edited to make it easier to follow and also the recent sponge culture study, Lopez-Acosta et a., 2018, has been included. The low Km simulation was already included in figure 6 and was/is referred to as Low Km and the Km value of the T. citrina species (Lopez-Acosta

2016) was/is given in table 2, Km T. citrina = 29.84 μM. For the low Km simulation, Km = 10 μM was chosen as a contrast with the published Km value, and because it was the Km value that fitted with the lower Δ³⁰Si limit of unfused sponges.

**P9 L26-31 –The argument presented here does not seem plausible or there is something missing in the text. How is efflux [rate] alone influenced by whether a bonding reaction is reversible or not? Can you provide some reasoning here? Even if the bonds are being created and/or destroyed simultaneously, would Si be removed from the organism? Further, the sentence on L29-31 is unclear. The Km of what organism? Or is this theoretical? Most organisms listed in the table have a Km that is much greater than 10 uM (i.e. 29.8-74.5 uM) therefore decreasing the Km to 10 uM doesn't seem likely, or am I missing something? Could you explain what you mean by 'the fractionation due to the efflux..'? Finally, please provide more information regarding the model that was presented as High E efflux and high Km. These are generally not very well described in the text.** The end of section 4.3 has been edited in order to present the efflux hypothesis with more details and explanations, now L18-30 p10.

**Problems with clarity**

**P1 L7 – what anomalies? Anomalous compared to what? Up until this point no anomalies have been described. Please include a sentence to describe what you mean by anomalies.** The anomalies represent the fractionation factor that fall outside of the calibration curve. We would like to keep the phrase "anomalies in the isotopic fractionation" as we believe that it concisely describes these data.

**P1 L8 – extremely light d30Si signatures? Compared to what? This was not described. Please clarify.** "compared with previous studies" L9p1 has been added.

**P1 L 10 – please clarify what you mean by spicule types. . .** "Spicule types" has been chosen here to describe the spicules shape/fusion without being too technical because we would like to avoid too many taxonomic words in the abstract.

**P1 L 15 – molecular fossil what? The molecular fossil record? Please clarify.** Done L16p1

**P1 L20 – Do you mean, Of the biomineralizing sponges? Please clarify.** Done, now L21p1

**P1/2 L21-2 – This sentence is unclear and I do not quite understand what the authors are trying to convey. This section needs to be developed a bit more and have a stronger link to the previous sentences so that I can understand why the authors wanted to include this information.** This sentence explains that the ratio of calcareous and siliceous sponges in the ocean has been changing with time due to the change in nutrient composition in the ocean. The end of the section has been changed to "but this ratio may have varied in the past due to changes in paleo-ocean chemistry (Montanez, 2002) because sponges rely on the ion chemistry of their surrounding water to build their skeleton" L2-3p2

**P2 L12 – Please provide more detail. Loose where within the skeletal framework? Can the authors please provide a better structural description here? Also what is meant by ". . .and they have a cellular organization." What kind of organization? Is it unique to each species? Also, the authors need to clarify that this sentence is discussing demosponge mega- and microscleres. As it is written, this is not clear in the text.** L12-15p2 has been edited: "which compose their skeletal structure. Either Mega- and microscleres are loose, unfused, but joined by spongin (collagen protein) (Uriz, 2006). Demospongiae have a cellular

organisation i.e composed of cells that form tissues, which themselves form organs, which form an organism."

**P2 L15 – what do you mean by rays? Spines along one of the three axes?** Now L18p2 The spicules are described with number of axes in the line before. The rays describe one of the "branches".

**P2 L16 – please clarify what you mean by secondary silica.** L20p2 "layer and or junction" has been added".

**P2 L17-18 – It is unclear as to why you have included this sentence. Please provide context and improve the conclusion of this paragraph.** The end of the paragraph has been edited, now L20-22p2 "One key feature that distinguishes between Demospongiae and Hexactinellida class is that Hexactinellida are characterised by a syncytial organisation, i.e. tissue composed of cells without individual plasma membrane (Leys and Lauzon, 1998; Maldonado and Riesgo, 2007)."

**P2 L23 – please clarify what you mean by 'sensitive to their environment'. How does growth rate and immobility make sponges sensitive to their environment? This point is unclear.** This sentence has been edited, now at L26-28p2 "Because of their relatively low growth rate and their immobility, they are sensitive to change of their environment and because an individual sponge can live decades or centuries (Pansini and Pronzato, 1990; Leys and Lauzon, 1998) they can record information over long time periods (Jochum et al., 2017)."

**P2 L 26 – this sentence needs to be reorganized – De La Rocha did not introduce the silicon isotopic composition of biogenic silica.** Done, now L30p2 "The silicon isotopic composition of biogenic silica ($\delta^{30}$BSi) has been introduced to study the past nutrient utilisation by 30 De La Rocha et al. (1997) and since has been used to study the silicon cycle (e.g Hendry et al., 2016; Fontorbe et al., 2017)."

**P3 L17 – the sentence '..by analysing d30Si along the sponge skeleton?' is not clear, please rephrase.** Done, now L20p3 "Can we trace DSi concentration over time by analysing $\delta^{30}$Si sections of sponge skeleton? "

**P3 L18-19 – This sentence is not clear. What precisely is being investigated? Please rephrase.** "$\delta^{30}$Si" has been added now L23p3

**P3 L22-25 – please rephrase this sentence, it is not clear.** Done, now L25-28p3 "Sponge samples were collected by remotely operated vehicle (ROV) and seawaters using Niskin bottles attached to CTD rosette system at five stations, EBA, EBB, VEM, VAY and GRM between 298 m and 2985 m (figure 1) aboard the RRS James Cook on the TROPICS cruise (JC094), a West-East cross section in the equatorial Atlantic between ~5°N and ~15°N, from the 13th October to the 30th November 2013. "

**P3 L26-27 – Please provide a list of the samples that were dried, preserved in ethanol and frozen. Please detail where exactly the samples were shipped in the UK.** "preserved in ethanol and frozen" has been deleted to not confuse the reader because none of the samples used in this study were preserved in ethanol or frozen (they were preserved by those means for other studies e.g. taxonomic identifications).

**P3 L28 – was the identification of the specimens to the major sponge classes carried out on subsamples that were preserved or were they fresh?** The major class identification was done on board (L3p4) based on analysis of the fresh sample.

**P3 L29-30 – what is the status of these identifications? Will the species ID be published as an appendix in this paper? Elsewhere? What journal?** L5p4 "in a separate paper" is added. The paper has not been submitted yet so we cannot give more details about it.

**P4 L7 – please rephrase the sentence 'If remaining, lithogenic material was removed by hand'. It is unclear.** The lithogenic material is visible by eye on living sponge spicules, we then remove it by hand if remaining. "before further cleaning steps" has been added L12p4 to precise that the spicule undergoes further cleaning steps.

**P4 L7 – Please clarify where the subsample originates. Is it cleaned? Has the lithogenic material been removed? This is not clear.** Done, L13p4 "A subsample was taken and weighed before going through a final cleaning step" was added.

**P4 L13 – Please clarify and rephrase 'Reynolds et al. (2006) modification.'** The MAGIC method is a common method used to analyse d30Si of seawater. The following sentence gives brief details of the method. This section has not been changed as other reviewers did not comment on it.

**P5 L7 – How was the D30Si calculated? There are no d30Si data for seawater presented in table A1.** In the corrected version d30Si of seawater have been added in table A1. It was not done in the first version because $\Delta^{30}Si = d30Si_{spicule} - d30Si_{seawater}$ is describe in the section 1.2 (now L9p3.)

**P5 L24-28 – This paragraph could be improved. It is not very descriptive and there is no flow. It reads more like a set of bullet points with, in some cases, poor grammar. Please explain why this information is important, for example, why has 'particular attention been paid to samples with a D30Si larger than 5 permil'? Do all samples show a common feature or just the samples that have a D30Si larger than 5 permil? The information is not abundantly clear from the text and needs to be clarified.** This paragraph has been entirely edited, see L3-8p6.

**P6 L3 – what studies? Please provide references** Done, see L11p6

**P6 L5 – please define epsilon f. what does it mean?** Sentence changed to "Here $\Delta^{30}Si$ is defined by the difference between $\delta^{30}Si_{Spicules}$ and $\delta^{30}Si_{DSi}$, which describes the observed apparent Si isotopic fractionation by sponges whereas $\varepsilon_f$ is the result from the biological model from Wille et al. (2010) (equation 2)" see L12-13p6

**P6 L26-27 – What do you mean by 'Despite the small range of temperature' – in this dataset? The Hendry and Robinson 2012 dataset? Please clarify.** Sentence changed, see L6-7p7 "Despite the small range of seawater temperature in this study, our data show no relationship between $\Delta^{30}Si$ and temperature (figure A1 in appendix)."

**P6 L29 – please explain what you mean by low concentration? What is the range?** The range was added, see L9p7

**P7 L4-7 – where are these data compiled? Please provide a reference or an appendix. Also, the authors need to be cautious about using the Wille et al. 2010 data set since they**

**did not measure the d30Si of seawater where their collected sponges resided. Their estimates for d30Si came from Cardinal et al. 2005. It would be a good idea to mention this in the text.** The data compiled for the residual a) are from the data presented in each paper from Hendry and Robison, 2012; Hendry et al., 2010; Wille et al., 2010. They are the same data presented in figure 2. The data from this study are detailed in table A1 with the fusion degree in the corrected version, which allow the reader to reproduce the residual. Because table A1 is very large, we decide to not add the data from previously published papers as they are already available. Wille et al., 2010 did not measure the d30Si of seawater but due to the conservative nature of d30Si in deep water masses the fractionation factor calculated in Willes et al., 2010 is valid.

**Minor revisions**

**Title: The authors do not provide any new information regarding the influence of biomineralisation on the silicon isotope composition of deep-sea sponges and I would recommend that they change the title to: "Silicon isotopes of deep-sea sponges: New insights from their skeletal structure"** We would like to keep the original title because despite that the study did not investigate the direct influence of the biomineralization on the silicon isotope but the relationship between the degree of fusion and the Kmp (half saturation constant of polymerisation) implies that there is.

**Plurals: please check over text for plural usage. Often, words are incorrectly pluralized. Please correct throughout the manuscript. Here are a few examples from the abstract:**

**P1 L9 – change 'insights' to 'insight', change 'process' to 'process(es)'** Done, now L9p1

**P1 L10 – change 'isotopes' to 'isotope'** Done, L10p1

**Definitions (e.g. Si, DSi, BSi ïA¸d'30Si, ïA¸e˘f): Please define these abbreviated terms correctly, and once defined, continue to use them instead of their non-abbreviated form. Check throughout the text.** Done throughout the text.

**Fractionation: Throughout the text the authors use the word fractionation but often do not describe what is being fractionated (e.g. silicon isotopes). Sometimes the word silicon fractionation is used, when the authors presumably mean silicon isotope fractionation. Please check throughout the text and correct this oversight. I have included a few examples below:**

**P1 L4 – add 'silicon' to 'apparent isotopic fractionation'**
**P1 L7 – add 'silicon' to 'isotopic fractionation'**
**P3 L5 – add 'silicon isotope' to 'apparent fractionation factor'**

**P3 L7 – add 'silicon isotope' to 'fractionation factor'**
**P3 L16 – chose to use Si or silicon (see section on definitions below)**
**P3 L16 – add 'silicon isotope' to 'fractionation'**
**P3 L18 – add 'silicon' to 'isotopic fractionation'**
**P5 L8– add 'silicon isotope' to 'fractionation'**
**P5 L21 – add 'silicon isotope' to 'fractionation'**
**P6 L5 – add 'apparent silicon isotope' to 'fractionation'**
**P6 L7 – add 'model' to 'Rayleigh-type fractionation' – also, change Raleigh to Rayleigh. P6 L9 – add 'isotope' to 'Si fractionation'**
**P6 L16 – add 'silicon isotope' to 'fractionation'**
**P6 L20 – add 'silicon isotope' to 'fractionation' (three times)**

**P6 L20 – add 'isotope' to 'Si fractionation'**
**P6 L29 – add 'silicon isotope' to 'fractionation'**
**P7 L20 – add 'isotope' to 'fractionation of Si'**

**P7 L4, P7 L27, P7 L31, P8 L3, P8 L5, P8 L19, P8 L24, P8 L25, P8 L30, P9 L3, P9 L8, P9 L9, P9 L20, P9 L21, P9 L30, P10 L2, P10 L9, Figure 2 caption, Figure 4 caption, Figure 6 caption, Figure 7 caption, Table A1. . .**

Because the paper is focused only on the silicon isotopic fractionation the "silicon isotopic fractionation" term was reduced in the previous version to have lighter sentence to read. This has been changed in the corrected version.

**P1 L1 – change to "The silicon isotope composition (d30Si) of deep-sea sponges' skeletal elements – spicules – reflect the . . ."** Done, L1p1

**P1 L18 – change to (Strehlow et al., 2010 and references therein)** Done, L19p1

**P2 L5 – Please change sentence to ". . ..spicules through the incorporation and deposition of hydrated amorphous silica (SiO2-nH20), otherwise known as bio-silica."** Sentence changed to "produce their spicules made of bio-silica (amorphous silica)" L4-5p2

**P2 L10-11 – These two sentences would be better merged.** New paragraph started L11p2 and the sentences have been merged.

**P2 L16 – change "loose" to "loosely attached"** We decided to keep "loose" to distinguish between cases where the spicules are attached to each other with secondary silica

**P2 L19 – please clarify what type of sponges (deep-sea/marine) – I think that you mean 'roused' and not 'aroused'. – also, it is the 'marine' silicon cycle.** Done, now L23p2

**P2 L21 – remove 'may be' and replace with 'are.** Now L25p2 has not been changed because this subject has been solely highlighted in one paper and the study used one area rather than a compilation to extend their standing Si stock to a global scale.

**P2 L 22 – remove 'immobility' and replace with 'inability to move'.** Done, now L26/27p2

**P2 L28 – please write 'approximately'.** Done, now L32p2

**P2 L29 – you need to include a statement about how the silicon isotope composition is expressed as permil. . . for example: '. . .are reported using delta notation as either d29Si or d30Si using the permil (ØL') scale. . .".** This has been changed to : "Silicon isotopic abundances in samples (SMP) are expressed as $\delta^{29}$Si or $\delta^{30}$Si with the abundance ratio, $^{29}$Si/$^{28}$Si or $^{30}$Si/$^{28}$Si respectively and measured relative to the reference standard (NBS28). The results presented in this study are expressed as permil to be consistent with the International Union of Pure and Applied Chemistry (IUPAC) nomenclature. i.e." (L33p2 to 2p3).

**P3 L3 – Please include Wille et al. 2010 and Hendry and Robinson 2012 here along with other references.** Done, now L6p3

**P3 L18 – replace 'issues' with 'questions' and 'are going to' with 'will'** Done, now L21p3

**P3 L22 – 'a remotely operated vehicle' and 'seawater was sampled using'** Done, now L25p3

**P3 L25 – please change to '. . .as smaller individuals encrusted on other organisms. . .'** Done, now L1p4

**P4 L3 – remove 'taken and', replace 'in' with 'into'** Done, now L8p4

**P4 L5 - replace 'in' with 'into'** Done, now L10p4

**P4 L7 - replace 'in' with 'with'** Done, now L12p4

**P4 L11 – please state the following sentence earlier in the paragraph: 'The cleaning procedure followed the technique in Hendry et al. (2010) and Hendry and Robinson (2012).** We decided to keep the cleaning procedure at the end of the paragraph. Now, L16p4

**P4 L22 – please add 'at the University of Bristol' 'after Bristol Isotope Group facilities'** Done, L28p4

**P4 L22 – change sentence to: '. . .were repeated at least twice..' and add the word 'methods' after 'Mg doping'** Done, L28p4

**P5 L 23 – please add 'marine' or 'deep-sea' P6 L8 – remove 'have'** deep-sea has been added, now L2p6

**P6 L9 – change to ' which suggests that silicon isotope fractionation in marine sponges is like to be controlled by a mechanism of Si uptake.'** "marine sponges" has been added but we kept "mainly". Now L17p6

**P6 L24-26 –remove 'concentration, supports Dsi concentration being the main factor controlling silicon isotope fractionation'. There still could be other factors such as pressure, salinity, etc.** This sentence has been kept as "which supports DSi concentration being the main factor controlling Si isotopic fractionation" (L6p7).

**P7 L21 – please move this information up to L13.** Done, now L25p7

**P7 L29 – remove 'the fact'** Done, now L9p8

**P8 L6-7 – 'A spicule is composed of hydrated amorphous silica (SiO2. . .' was already defined on P2 L5. The purpose of this sentence is unclear, please rephrase.** This sentence introduces the reader to this new section that focus on the spicule composition, so this information is a key point. This sentence is a reminder like that reader does not need to return to the beginning of the paper to understand the rest of the paragraph. "and organic molecules (Uriz et al., 2003)" has been added at the end of the sentence. Now L21p8

**P8 L7 – remove 'The' as in "The biosilicification'** Done, now L23p8

**P8 L 21 – change to 'sponge E. aspergillum is comprised of small spicules that are embedded in a silica matrix surrounding a larger** Done, now L5p9

**P8 L26 – change to '. . .solely a result of the differences in organic** composition' Done, now L9-10p9

**P8 L 30 – change to '. . .Si isotopes by sponges, epsilon f (see equation 2).** Done, now L13p9

**P8 L32 – please provide a reference for your definition of efflux (Milligan? Wille? Other?).** "(Wille et al., 2010, and references therein)" has been added, now L16p9

**P9 L1-2 – please consider changing to 'values from the aforementioned studies in four different laboratory-based sponge culture experiments (summarized in Table 2).'. Remove 'and with KmP and Vmaxp, the maxium polymersation rates.'** The end of the paragraph has been changed to "To date, only Reincke and Barthel (1997); Maldonado et al. (2011); López-Acosta et al. (2016) and López-Acosta et al. (2018) have cultured sponges to investigate the 15 Michaelis-Menten enzyme kinetics of sponges. In this section, $\varepsilon_f$ has been modelled using $K_{m,P}$ and $V_{max,p}$ values from these four sponge culture experiments and are summarised in table 2." L16-19p9

**P9 L24 – change to ' Biosilicification in sponges results in the condensation. . .'. Enzyme should be plural.** Done, now L18p10

**Figure 2 caption – please add the abbreviated terms d30Si, D30Si and Si(OH)4 to the figure caption.** Done, "a) Silicon isotopic composition of the spicules ($\delta^{30}Si_{Spicules}$) and b) deep sea sponges apparent Si isotopic fractionation ($\Delta^{30}Si$) against DSi" has been added.

**Figure 7 – please clarify in the caption that these data are only from the current study.** Done, "from this study" has been added.

**Table 2 – please define the parameters listed in the table in the table caption. Capitalize the first letter of 'reference'** Done, "with$V_{max,p}$ the the maximum polymerisation rates and $K_{m,P}$ the half saturation constant of polymerisation." Has been added.

Response to Referee #2

**One issue that could be easily improved is the presentation of the data. Currently, it is not possible to recreate the authors analysis because the spicule fusion degree isn't given in the data tables – this could be easily remedied, and it should be also possible to code the symbols in the figures so one can see where the different fusion levels fall.** The data table A1 has been edited and the fusion degree of the spicules is now available. We have change figure 2 , which now has the fusion degree included, but we did not do this for figure 5 and 6 because figure 5 focuses on the comparison between Hexactinellids and Demosponges, and figure 6 is already very colourful due to the 8 $\varepsilon_f$ simulations. Adding additional colour-coding for the degree of fusion would make both figures  unreadable.

**More generally, because it was not possible to plot the data myself, I became a bit confused at parts, regarding what the takeaway message should be. Some samples fall off all the versions of the fractionation concentration regression: P5L20 makes clear that this is related to the degree of spicule fusion, which I interpreted from the introduction in general and P6L28 & P7L2 specifically to be related to the taxonomy of the samples, specifically whether they were hexactinellid or demosponges. But then P8L24 and the residual tests says this is not the case. It seems the deviation from the 'average' sponge is the most useful indicator, which is what the residual plots are showing – but in the end it's unclear whether or not this is related to fusion degree or to the taxonomy.**  Previously L28p6 "a group of hexactinellid sponge" has been changed to "a group of sponge" now L8p7 to avoid confusion. **My feeling is that if Fig. 4 was altered to show some measure of**

**deviation/residual, rather than absolute values, and the discussion altered to reflect this, things would be clearer. Otherwise surely the default interpretation of Fig. 4 is that different taxa like to live in different parts of the ocean?** Figure 4 was only representing the samples identified from this study, which come from the deep-water masses of the equatorial Atlantic. This information has been added in the caption to avoid confusion. Furthermore, in the corrected version, figure 4b is added and shows the fractionation factor residual of the 5 degrees of fusion compared to the published calibration curve. **Similarly, I stuggled to follow the rationale for the discussion in section 4.3 – it seems trivial if you have 5 (?) tunable parameters, you can make a model produce any magnitude of fractionation. Or have I missed the point here? I would recommend trying to emphasise the key point.** Section 4.3 has been entirely edited and is now organised in 2 parts: first the Kmp is investigated and then the efflux. Kmp is first investigated due to the relationship between εf simulations results and the Michaelis-Menten parameters and then pushed to extreme value, Kmp=10µM, to see the effect on εf. Because εf of the dictional skeleton cannot be simulated by Kmp we then investigate the effect of the fractionation of the efflux. The section is now better structured to help the reader.

**Finally, I wonder if the authors have given any thought to whether this difference in Si-isotope fractionating behaviour between different sponge types is something that could be exploited rather than avoided in a paleoceanographic context?** At the end of section 4.3 we suggest that Si isotopes could potentially be used to investigate cellular uptake and silicification processes, see L29-30p10.

**Minor comments - this is a non-exhaustive list of small typographic, etc. errors and some comments/questions.**

**P1L5: "ranges"** Done, now L5p1

**P1L15: "fossils"** Now "fossil records" L16p1

**P1L18: "and references therein"** Done, now L19p1

**P1L20: "Since sponges rely" or "because sponges rely"** Done, now L2p2

**P2L5: either "from" or "of", not both** Done, change to ''made of'', now L5p2

**P2L9: An approximate threshold size for distinguishing between micro- and megascleres would be useful** Size has been added "megascleres (up to and beyond 300 µm) and microscleres (up to 50 µm)", now L9p2

**A reference to Jochum et al. (2017) might be appropriate here P2L29:** Done, now L28p2

**Phrasing is unclear, P2 Eqn 1: the permil is not necessary.** We haven't changed it because x1000 is not included into the equation the permil symbol show that the delta value is express in permil unit.

**P3L9: Reference to Fontorbe et al. (2016) and/or Fontorbe et al. (2017) might be appropriate here** Reference Fontorbe et al., 2017 has been added, now L13p3

**P3L11 "result"** Done, now L14p3

**P4L5: "to", not "in"** change ''to'' by ''into'', now L10p4

**P4L7: How do you know all the lithogenic material was removed? Could it be contaminating the samples?** The sponge spicules analysed here are from live sponges. The potential lithogenic material is visible by eyes if remaining after the first steps of cleaning, and it is possible to remove it. Furthermore, further cleaning steps are done with high concentrated acid, which results in very clean  spicules, so we are confident that the spicules are free of lithogenics. The sentence has been edited to "If remaining, lithogenic material was removed by hand before further cleaning steps. A subsample was taken and weighed before going through a final cleaning step'', see L12-13p4

 **P4L12: "prior to isotopic", and "induced"** Done, L18p4

**P4L15: Has it been tested that the yield, including the washing step, is quantitative?** Yes, the yield was tested and is now included in the method, see L23p4 "The yield recovery of Si is equivalent to 92.1%"

**P4: Were any seawater standards analysed? (AHOLA, from Grasse et al. (2017)).** We have analysed the ALOHA Deep water standard using this method and have added the results, see L1-3p5 '' The new seawater standard ALOHA deep was analysed as an additional quality check, and yielded values within error of those obtained during an interlaboratory (Grasse et al., 2017): Aloha deep: $\delta^{30}$Si = 1.08± 0.12 ‰ and $\delta^{29}$Si = 0.58± 0.12 ‰ (2 s.d, n = 4).''

**P4L21: "spectrometry"** Done, Now L27p4

**P5L20: The sentence starting "the d30Si_spicules. . ." is oddly phrases – it could just say 'd30Si_spicules and apparent fractionation both increase. . .'** The end of the paragraph has been changed to '' It is observed that $\delta^{30}$Si$_{Spicules}$ and $\Delta^{30}$Si show an enrichment in light isotopes with higher degree of spicule fusion (se figure 4).'' Now L27-28p5

**P6L5: Not sure it's correct to say epsilon can result only from a biological model – epsilon is simply the permil expression of fractionation factor alpha – see Coplen 2011 "Guidelines and recommended terms for expression of stable- isotope-ratio and gas-ratio measurement results"** We have changed the sentence to "Here $\Delta^{30}$Si is defined by the difference between $\delta^{30}$Si$_{Spicules}$ and $\delta^{30}$Si$_{DSi}$, which describes the observed apparent Si isotopic fractionation by sponges whereas $\varepsilon f$ is the result from the biological model from Wille et al. (2010) (equation 2)" (L12-14p6) to distinguish the observe and the calculated fractionation factor, which follow the previous published paper.

**P6L7: I don't see the relevance of mentioning Rayleigh fractionation here, these are samples from all over the ocean, not a single site evolving through time.** This sentence is to emphasize that sponge Si isotopic fractionation does not follow the Raleigh type fractionation observed in diatoms.

**P7L1: Or, more generally, that different taxa have different epsilons, Km or Vmax – worth mentioning here?** The end of the paragraph has been changed but not the last sentences.

**P7L10: "in different ways"** Done, now L22p7

**P7L18: "from Hendry and Robinson"** Done, now L30p7

**P7L33: "main"** Done, L15p8

**P8L4: "impact the fractionation".** Done, L19p8 **Also, a bit of skepticism about the ab initio calculations might be warranted.**

**P8L17/18: Is there an indication of uncertainty on the 'thermal analysis' – are 10% and 15% really different?** The uncertainties on those values are not known, to our knowledge but to support it another experiment and reference is added, see L33p8 L2p9 "Furthermore, SDS-PAGE analysis of Hexactinellid *Euplectella aspergilum* has shown that the proteins of its axial filament have higher molecular weights than those isolated in demosponges (Weaver and Morse, 2003),"

**P8L30 (and elsewhere): 'Sponge fractionation' would more correctly read as 'fractionation of silicon isotopes by sponges' or something similar.** Done

**P8L31: This sentence is missing a verb.** The sentence has been changed to '' The Si isotopic fractionation by sponges is assumed to occur during Si uptake and during internal spicule formation with spicule formation being a function of Si influx and efflux from the sclerocyte (Milligan et al., 2004)'', see L13-14p9

**P9L6: "described"** Done, now L24p9

**P9L24: "enzymes"** Done, now L23p10

**P9L28: "breaking of bonds"** Done, now L22p10

**Figures and Tables Figure 1: I guess the colours represent bathymetry – a color bar would be nice, and perhaps another panel showing a cross-section of dissolved silicon concentrations along the sampling transect.** Colour bar for the bathymetry and DSi cross section have been added.

**Figure 2, 5 and 6: It would be convenient to show the different spicule types, perhaps color coded somehow, as in figure 7.** The colour coding is mentioned and commented in general comments. **I also notice that an outlier from Hendry and Robinson isn't shown – would recommend mentioning this somewhere to be clear.** In fact, the core top samples from Hendry and Robinson (2012) are not included. This is now mentioned in figure 2 caption.

**Figure 4: Why are the individual data points only shown for spicule fusion degree F5?** Data points for all the fusion degree have been added. In the first version, only F5 was detailed to show why the error bars was so large. **More generally, wouldn't it be more useful to plot the residuals as discussed in the main text?** A residual test is now added, see figure 4b **Otherwise all this figure could be telling us is that sponges that produce highly fused spicules prefer to live in the deep sea/high Si concentration waters (and Alvarez et al. (2017) recently showed that different groups do seem to have distinct depth preferences). See also comments above.** Figure 4a only show the samples with identified fusion degree from the deep water of the equatorial Atlantic indicating that there is a significant relationship between the Si isotopic fractionation and the degree of fusion. We have added the following to the caption of figure 4: '' Data plotted here correspond to the samples from the equatorial Atlantic with identified fusion stage i.e. coloured diamond of b) and so occupy a very narrow range of DSi. b) $\Delta^{30}$Si residual of samples with identified fusion stage compared to the published calibration (best fit 1).''

**Figure 5: Would it make sense to plot the residuals for all data for each possible regression? i.e. then one could see where the new data sits with respect to the previously**

**published data.** Figure 4b has been added and show the residual test of the fusion degree compared to the previous published data. **Also, it would good to see a justification for an expression of this form being chosen rather than e.g. linear fits, power laws, etc.** The best fit equations are following the fractionation factor equation (equation 2), according to published methodology (e.g. Wille et al., 2010; Hendry & Robinson, 2012).

**Figure 7 caption: Atlantic.** Done
**Table 2: The recent data from López-Acosta et al. (2018) could be incorporated here.** The recent data from Lopez-Acosta et al., 2018 are now included.

**Figure A1: Could the global data compilation also be presented here?** The temperature data of all data compilation is not available, **the caption, 'ambient', not 'ambiant'.** Done

**Table A1: For this work to be most useful/reproducible, this table should include the class of fusion degree each sample has been assigned to. I would have like to plot the data myself but was unable to.** The fusion degrees are now included into table A1.

Response to referee #3

**I agree with reviewer 1 that a better quantification of the levels of spicule fusion would be desirable. Also an identification of the level of fusion within the table A1 in the appendix would be nice with a possible database of the SEM images for each sample. This enables the reader to get an idea of changing spicule morphology and how it is related to the apparent Si isotopic fractionation.** The degree of fusion has been added into table A1. However, we did not add pictures because the table is already very large, and examples of the various degrees of fusion are given in figure 3. It is mentioned in the sample availability section that images are available upon request. **This is one important aspect since from the residual plot in Figure 5d it can be seen that compared to the Demospongiae, Hexactinellida exhibit not only the lightest, but the heaviest D30Si residuals (also seen in Figure 4, although mean and median of all sponges in this group are lighter).**

**Here sponges (Hexactinellida and Demospongiae) from the location GRM 17.4306 53.1831 at 1869m show heavy values. What do spicule morphologies of these samples look like how is their degree of fusion?** The fusion degree of the samples from GRM with heavy d30Sispicule are now documented in table A1 and show a F1 and F5 degree of fusion. **Sadly I was not able to find the sample with the second highest D30Si (figure 5c at >30uM Si(OH)4. From figure 5d it seems to be a Hexactinellida. Saying this, there is a mismatch between figure 5c and Figure 6 although both figures present all published data and all JC094 data. Also please also compare if all data presented in Figure 5c are also displayed in the residual plot 5d.** There is a mismatch between figure 2, 5c and 6 because not all samples have been identified. We have clarified this point in figure 5 caption to avoid confusion. **Some question regarding the dissolved Si concentrations: why are the dissolved Si concentrations for e.g. samples of the GRM location always 15.96 uM although they have a distance of more than 250 km and a depth difference of 400m?** Unfortunately, it was not always possible to collect co-located sponge and water samples: the water sample closed to the sponge location was analysed. **Can these sites, close to the mid Atlantic ridge, are affected by dissolved Si supplied by hydrothermal vents? How such a supply, although temporally, might influence the morphology of the spicules? Hydrothermal waters enriched in silica seems to promote the development of sponge growth with have thicker spicules compared to sponges that live in Si poor waters?** We have not discussed the potential effect

of hydrothermal supply. Sampling of waters directly above the TAG hydrothermal site did not reveal any anomalies in dissolved Si concentrations and only a very local anomaly in silicon isotopic composition of seawater (Brzezinski & Jones, 2015; Deep Sea Research Part II: Topical Studies in Oceanography volume 116); we did not sample any sponges or waters sufficiently close to any active sites to their to have been an impact on sponge growth or isotopic composition.

**After excluding such external, environmental parameters section 4.2 and 4.3 discuss the effect of organosilicon complexes on Si isotopic fractionation. Within section 4.2 conclude that the absent "difference in the fractionation between Hexactinellida and Demospongiae classes despite the difference in their spicule composition, suggesting that the large fractionation in sponges that display a dictyonal framework is not solely a result of the organic composition of the spicules but could be controlled by the enzymes that me- diate silica deposition." However only Hexactinellida show a higher degree of fusion (figure 1). So why can the development of a silicon matrix or fusion of Si, possibly with the help of organosilicon complexes, can not be a additional Si deposition method for Hexactinellida?** The end of section 4.2 has been changed to '' This suggests that the large $\Delta^{30}$Si in sponges that display a dictyonal framework is not solely a result of the differences in organic composition of the spicules but could also be controlled by the enzymes that mediate silica deposition."

**I think that the model calculation in 4.3 is valid since differences in Si uptake parameters are between sponges are obvious and will have influence on the apparent Si isotopic fractionation. But expect one very light data at   125 uM Si(OH)4 (Hendry & Robinson, 2012) Hexactinellida and Demospongiae from compiled previous publications from different oceans do show a relative small range in residual D30Si. I admire that this manuscript double the amount of available sponge Si isotope data, but if uptake parameters between sponges such a large effect on apparent Si isotope fractionation, why it has not been discovered it in the compiled previous datasets? Does this give indication that environmental parameters at some sample location presented here are different compared to previous publications?** An entire section has been added

 "4.4 Comparison with previous studies

The new data set of this study show a wide range of $\Delta^{30}$Si for a small range of DSi concentration compared to previous studies (figure 2) (Hendry and Robinson, 2012; Wille et al., 2010; Hendry et al., 2010). Here spicule shape, in particular the fusion stage, and $\Delta^{30}$Si have been investigated and are closely related, where high fusion stages show very large $\Delta^{30}$Si, which deviate from the existing calibration. Why has this relationship between spicule fusion and $\Delta^{30}$Si not been observed in previous studies? This new data set is composed of 103 samples in which 15 are deviating from the calibration and display a dyctional skeleton. Previous studies are based on fewer samples and all the hexactinellid specimens have been found, except for one, in high DSi environments (higher than 45 µM) (Hendry and Robinson, 2012; Wille et al., 2010; Hendry et al., 2010). As the spicules in Hexactinellida class can be loose, partially or totally fused, or even cemented by secondary silica (Uriz et al., 2003), it is likely that previous studies only analysed samples with loose  spicules (equivalent to F1 here). Furthermore, Hendry and Robinson (2012) piblished one sample with $\Delta^{30}$Si = -6.52 ‰ for DSi = 125 µM (figure 2). This sample also displayed a fused skeleton but at this date the large fractionation was attributed to the lack of constraint on ambient seawater $\delta^{30}$Si ."

---

## Referee Report (RR1)

This authors have done a good job at addressing most of reviewers comments and all of the major criticisms appear to have been addressed correctly. I note that a few of the reviewers' comments were dismissed as trivial, but I think these 'minor' comments should be reconsidered. For example:

(1) **Reviewer 1, - P7 L4-7:** This reviewer indicates that there is not a reference to where the data are compiled (presumably within the manuscript). The authors respond that the data are presented in published papers (which are included in the manuscript) and Figure 2 (not referenced here). *I believe that the reviewer is simply requesting that the authors refer to Figure 2* on P7 L4-7.

(2) **Reviewer 2, - P6L7:** This reviewer does not see the relevance of presenting Rayleigh-type fractionation. The authors respond that this is to emphasize that sponge Si isotopic fractionation does not follow Raleigh (should be spelled Rayleigh) type fractionation observed in diatoms. Up until this point, the authors do present anything related to diatoms or Rayleigh-type fractionation, and thus the comment appears to be out of context. *I recommend that the authors either remove this comment about Rayleigh-type fractionation or provide a better argument within the text as to why it has been presented (e.g. explain the Rayleigh distillation model and why it is important for diatoms and paleoceanography).*

(3) **Reviewer 3:** This reviewer wrote "Some question regarding the dissolved Si concentrations: why are the dissolved Si concentrations for e.g. samples of the GRM location always 15.96 uM although they have a distance of more than 250 km and a depth difference of 400m?" The authors responded: "Unfortunately, it was not always possible to collect co-located sponge and water samples: the water sample closed to the sponge location was analysed." *I think it would be good to mention this in the text under section 2.1 – Sample collection.*

---

## Author Response (AR2)

**Response to report 2 – Minor revisions**

Each comment has been taken into consideration and has been corrected. Please find below the response to each comment with in bleu and bold the reviewer comment and the response in black. We would like to thank again the reviewers for their time and constructive comments.

(1) **Reviewer1,-P7L4-7: This reviewer indicates that here is not a reference to where the data are compiled (presumably within the manuscript). The authors respond that the data are presented in published papers (which are included in the manuscript) and Figure 2 (not referenced here). I believe that the reviewer is simply requesting that the authors refer to Figure 2 on P7 L4-7.** Figure 2 is now referred. See P7 L 23-24 "The $\delta^{30}Si$Spicules average signature of the two siliceous sponge families from the compiled data presented in figure 2 (Hendry and Robinson (2012), Wille et al. (2010), Hendry et al. (2010) with the equatorial Atlantic data (JC094)) show that the Hexactinellida class is significantly lighter than the Demospongiae, with $\delta^{30}Si$Spicules = −2.66 ± 0.21 ‰ (C.I. of the mean) and −1.91 ± 0.30 ‰ (C.I. of mean) respectively.

(2) **Reviewer2,-P6L7: This reviewer does not see the relevance of presenting Rayleigh-type fractionation. The authors respond that this is to emphasize that sponge Si isotopic fractionation does not follow Raleigh (should be spelled Rayleigh) type fractionation observed in diatoms. Up until this point, the authors do present anything related to diatoms or Rayleigh-type fractionation, and thus the comment appears to be out of context. I recommend that the authors either remove this comment about Rayleigh-type fractionation or provide a better argument within the text as to why it has been presented (e.g. explain the Rayleigh distillation model and why it is important for diatoms and paleoceanography).** The comment has been kept into the text with further argument about the Rayleigh type fractionation in order to clarify the sentence. See P6 L 17-19 "Published data have shown $\Delta^{30}Si$ varying from -0.77 ‰ to -6.52 ‰ (figure 2b), which follow a non-linear relationship and cannot be described by a diatom-like Rayleigh fractionation (characterised by a constant fractionation factor during DSi utilisation) because isotopic fractionation during the uptake of DSi by sponges is variable, increasing with DSi concentration."

(3) **Reviewer3: This reviewer wrote "Some questions regarding the dissolved Si concentrations: why are the dissolved Si concentrations for e.g. samples of the GRM location always 15.96 uM although they have a distance of more than 250 km and a depth difference of 400m?" The authors responded: "Unfortunately, it was not always possible to collect co-located sponge and water samples: the water sample closed to the sponge location was analysed." I think it would be good to mention this in the text under section 2.1 – Sample collection.** This information has been added into the section 2.1, see P3 L 26 "Sponge samples were collected by remotely operated vehicle (ROV) at five stations, EBA, EBB, VEM, VAY and GRM be- tween 298 m and 2985 m (figure 1) aboard the RRS James Cook on the TROPICS cruise (JC094), a West-East cross section in the equatorial Atlantic between ∼5°N and ∼15°N, from the 13th October to the 30th

November 2013. Seawater was sampled using Niskin bottles attached to CTD rosette system and occasionally by ROV at each station. Whilst best attempts were made to spatially match the sponge and water samples, it was not always possible to collect precisely co-located sponge and seawater samples. The $\delta^{30}Si_{DSi}$ values are reported in table A1 (appendix) and, for each sponge specimen, the closest seawater sample is used to calculate $\Delta^{30}Si$.

---

## Author Response (AR3)

**Response to Associate editor minor corrections:**

- **P. 4, l.5, why a hard-return at the end?** The hard-end has been removed. The hard-return at the end of the sentence was to separate storage method the and the sponge specification as they are to different and singular steps in the method.
- **P. 4, l.24, please replace rpm with g-force. Rpm have no meaning without specification of radius, etc** rpm have been changed to g-force. Now p.4, l.23
- **P. 5, l.7, … during an interlaboratory evaluation/assessment/study?** Study has been added, now P.5, l.6.
- **P. 6, l.15: The residual test, however, highlights…. (on the one and on the other hand always go together).** Did you mean P. 8, l.15? If yes, "on the other hand'' has been changed with "however", now P.8, l.15.
- **P. 9, l.21-22: The sentence as proposed … (see equation 2) does not make sense. Something missing here.** The sentence has been changed to "As proposed by Wille et al. (2010) the fractionation of Si isotopes by sponges, $\varepsilon f$ , can be expressed by equation 2", now P. 9, l.21-22.
- **P.11, l.14: published.** Piblished has been changed to "published", now P.11, l.14.